# Seasonal evolution of the supraglacial drainage network at Humboldt Glacier, North Greenland, between 2016 and 2020

Lauren D. Rawlins[1*], David M. Rippin[1], Andrew J. Sole[2], Stephen J. Livingstone[2], Kang Yang[3]

[1]Department of Environment and Geography, University of York, YO10 5NG, UK,
[2]Department of Geography, University of Sheffield, S3 7ND, UK,
[3]School of Geography and Ocean Science, Nanjing University, Nanjing, People's Republic of China

*Correspondence to*: Lauren D. Rawlins (lauren.rawlins@york.ac.uk)

**Abstract.** Supraglacial rivers and lakes are important for the routing and storage of surface meltwater during the summer melt season across the Greenland Ice Sheet (GrIS), yet remain poorly mapped and quantified across the northern part of the ice sheet, which is rapidly losing mass. Here we produce, for the first time, a high-resolution record of the supraglacial drainage network (including both rivers and lakes) and its seasonal behaviour at Humboldt Glacier, a wide-outlet glacier draining a large melt-prone hydrologic catchment (13,488 km$^2$), spanning the period 2016 to 2020 using 10 m spatial resolution Sentinel-2 imagery. Our results reveal a perennially extensive yet interannually-variable supraglacial network extending from an elevation of 200 m a.s.l to a maximum of ~1440 m a.s.l recorded in 2020, with limited development of the network observed in the low melt years of 2017 and 2018. The supraglacial drainage network is shown to cover an area ranging between 966 km$^2$ (2018) and 1566 km$^2$ (2019) at its maximum seasonal extent, with spatial coverage of up to 2685 km$^2$ recorded during the early phases of the melt season when a slush zone is most prominent. Up-glacier expansion and the development of an efficient supraglacial drainage network as surface runoff increases and the snowline retreats is clearly visible. Preconditioning of the ice surface following a high melt year is also observed, with an extreme and long-lasting 2019 melt season, over-winter persistence of liquid lakes followed by low snow accumulation the following spring culminating in earlier, widespread exposure of the supraglacial drainage network in 2020 compared to other years. This preconditioning is predicted to become more common with persistent warmer years into the future. Overall, this study provides evidence of a persistent, yet dynamic, supraglacial drainage network at this prominent northern GrIS outlet glacier and advances our understanding of such hydrologic processes, particularly under ongoing climatic warming and enhanced runoff.

# 1 Introduction

The Greenland Ice Sheet (GrIS) has experienced significant mass loss throughout the 21$^{st}$ Century and currently represents the largest single cryospheric component of global sea level rise, contributing an estimated $10.6 \pm 0.9$ mm since 1992 (IMBIE, 2020). Over the last two decades, GrIS mass loss has become increasingly dominated by surface mass balance (SMB) processes, accounting for 60% of ice loss annually since 1991, with the remainder attributed to dynamical mass losses from marine-terminating glaciers along the ice sheet periphery (van den Broeke et al., 2016). Such SMB losses are being increasingly revealed by the magnitude and spatial extent of seasonal surface melting and runoff (Trusel et al., 2018), attributed to climate-driven atmospheric warming (Hanna et al., 2012; Hanna et al., 2021), summertime atmospheric circulatory behaviour (i.e. Greenland Blocking Index; Hanna et al., 2012; 2021; McLeod et al., 2016; van den Broeke, 2017) and the ongoing expansion (Noël et al., 2019) and darkening (Tedesco et al., 2016; Ryan et al., 2018; 2019; Riihelä et al., 2019) of the bare ice zone. Between 2011 and 2020, runoff was 21% higher than any of the preceding three decades (Slater et al., 2021).

Surface runoff is transported by an expansive and complex supraglacial drainage system which is activated during the summer season (Pitcher and Smith, 2019). This drainage system, made up of an ephemeral network of interconnected supraglacial rivers and lakes, transports and stores large volumes of surface meltwater on the ablating ice surface (Rippin and Rawlins, 2021). Such runoff can become intercepted by crevasses and moulins, that provide connections to the ice sheet bed where the timing and delivery of such water has been shown to affect ice velocity (Zwally et al., 2002; Bartholomew et al., 2010; 2012; Hoffman et al., 2011; Sole et al., 2011; Andrews et al., 2014; Nienow et al., 2017). In particular, meltwater delivery into an inefficient subglacial configuration, such as linked cavities (Kamb, 1987) which typically occurs during the early period of the melt season, can temporarily overwhelm the subglacial hydrologic system, increasing water pressure and enhancing subsequent sliding (Andrews et al., 2014; Davidson et al., 2019). In some regions where moulins and crevasses are absent, supraglacial rivers can extend undisturbed for tens of kilometres across the bare ice surface, flowing directly into the proglacial zone (Yang et al., 2019a; Li et al., 2022). Ultimately, much of this meltwater will end up in the ocean, contributing directly to global sea level rise (Pitcher and Smith, 2019).

Whilst many remote sensing studies have examined components of the supraglacial drainage network in-depth across the largest melt producing western and southwestern sections of the GrIS (Smith et al., 2015; Gleason et al., 2016, 2021; Yang et al., 2021), it is only recently that other significant ice sheet sectors have begun to be mapped (Gledhill and Williamson, 2018; Macdonald et al., 2018; Yang et al., 2019a; Schröder et al., 2020; Turton et al., 2021; Lu et al., 2021; Bogshosian et al., 2023). Focus has only recently shifted to the rapidly changing northern regions of the GrIS, with evidence of inland expansion of supraglacial lakes observed in north east Greenland (Turton et al., 2021) which align with climate model projections (Leeson et al., 2015) and the existence of a widespread supraglacial network (Lu et al., 2021). This study utilises Sentinel-2 satellite imagery to map the supraglacial drainage network, including both rivers and lakes, on a major northern outlet glacier of the

GrIS - Humboldt Glacier (79°23.86°N, 64°20.60°W), hereby denoted to HG – to examine its seasonal behaviour at high spatial
(10 m) and temporal resolution over five consecutive melt years (2016 – 2020).
**2 Study Location**
The drainage basins of outlet glaciers in the northern sector of the GrIS comprise ~14% of the total ice sheet area, with 82%
of the northern sector predominately drained by 12 marine-terminating glaciers which together hold a sea level equivalent of
93 cm (Mouginot et al., 2019). Since 1990, this sector has experienced some of the most pronounced changes in surface melt
and runoff, attributed to the rapid expansion of the ablation (46%) and bare ice (33%) zone at rates twice as fast than in southern
Greenland, with this trend expected to continue with ongoing climatic warming (Noël et al., 2019). Of the northern outlet
glaciers that drain the GrIS, HG, also known as Sermersuaq Glacier, is the widest marine-terminating outlet glacier (~91 km
wide) in Greenland and is responsible for draining ~5% of the ice sheet alone north-westward into the Kane Basin (Hill et al.,
2017; Rignot and Kanagaratnam, 2006; Rignot et al., 2021; Fig. 1). Since the late 1990s, HG has experienced rapid rates of
retreat (~162 m a$^{-1}$) attributed to increases in mean summer air temperatures and sea-ice decline (Carr et al., 2015). Holding
an ice volume equivalent of 19 cm of sea level rise, HG is the fourth largest Greenland glacial contributor to sea level rise
(Rignot et al., 2021) having lost 161 gigatonnes (Gt) since 1972 and 311 km$^2$ of its area between 2000 and 2010 (Box and
Decker, 2011).

Until recently, few glaciological studies had focussed on HG (Joughin, et al., 1996, 1999; Carr et al., 2015; Livingstone et al.,
2017; Hill et al., 2017, 2018; Mouginot et al., 2019; Gray, 2021; Rignot et al., 2001, 2021, Hillebrand et al., 2022). Studies
that have examined HG identified a distinctive ice velocity divide between the northern and southern sectors (Rignot et al.,
2001, 2021; Carr et al., 2015; Fig. 1d); the northern sector has up to four times faster ice flow than the south. In terms of
surface hydrology, several studies have noted the presence of supraglacial lakes (SGLs; Joughin et al., 1996; Selmes et al.,
2011; Carr et al., 2015), however none to-date have examined the overall drainage system, including both rivers and lakes, in-
detail.









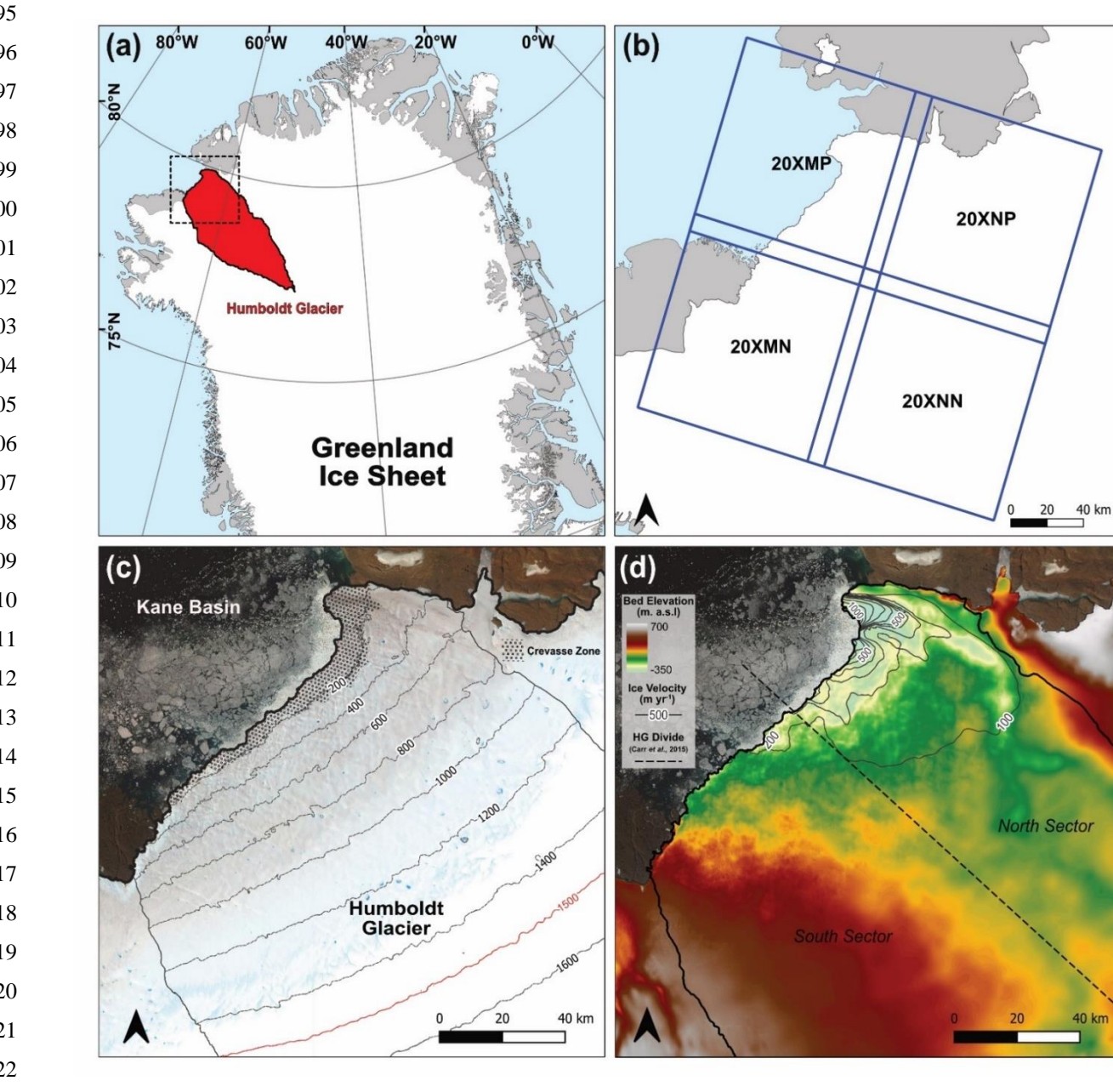

**Figure 1. (a) Study location of HG, north Greenland, and its highlighted drainage basin (shaded red). The dashed box shows the inset for other figure boxes (b-d); (b) The four Sentinel-2 tiles used for extraction of the supraglacial drainage network across the study region; (c) True colour (R: band 4; G: band 3; B: band 2) Sentinel-2 image of HG acquired from 25th July 2020 courtesy of the Copernicus Open Access Hub (https://scihub.copernicus.eu). HG denoted drainage basin and 200 m contour lines derived from ArcticDEM (10 m) are shown. The 1500 m a.s.l contour denotes the maximum melt extent. The shaded section up to 200 m a.s.l also shows the heavily crevassed zone that exists within the northern sector of the terminus; (d) Bed topography of HG and the surrounding area from Bed Machine version 4 (Morlighem et al., 2021), ice velocity contours via NASAs MEaSUREs ITS_LIVE project (Gardener et al., 2019) and the division of Humboldt's north and south sectors (dashed line) as per Carr et al. (2015).**

## 3 Data and Methodology

### 3.1 Data sources

Earth-observing satellites enable the study of supraglacial drainage features with broad spatial and temporal coverages (Rennermalm et al., 2013; Yang and Smith, 2012; Chu, 2014). Over the last four decades, the Landsat program has provided a wealth of remotely-sensed data for the mapping and quantification of a number of supraglacial features such as SGLs (Lampkin and Vanderberg, 2011; Banwell et al., 2014; Pope et al., 2016; Williamson et al., 2017; Gledhill and Williamson, 2018; Williamson et al., 2018; Yang et al., 2019b; Dell et al., 2022; Otto et al., 2022), as well as for exploring the generalised configuration (i.e., main river stems) of the supraglacial drainage system (Lampkin and Vanderberg, 2014; Yang et al., 2021). But, its spatial resolution in the visible spectrum (30 m) precludes the reliable delineation of numerous smaller supraglacial rivers (Yang et al., 2019a). This has resulted in these complex networks being unmapped and underrepresented (Chu, 2014). The application of the Multispectral Instrument (MSI) on Sentinel-2 satellites (Sentinel-2A and -2B), which launched in 2015 and 2017 respectively, offers a higher-resolution (10 m) perspective of such systems (Yang et al., 2019a; Lu et al., 2020; Lu et al., 2021). Sentinel-2 imagery enables the detection and delineation of both wide, main-stem river channels, which have high stream orders and are perennially reoccupied (Pitcher and Smith, 2019) as well as narrower (one pixel, or 10 m), tributary-style channels that are lower-order and shallower in depth (Smith et al., 2015; Fig. 2). Sentinel-2 imagery has also been shown to better-resolve supraglacial networks in general for mapping purposes at a glacier-wide scale, particularly in terms of river continuity (Yang et al., 2019a), hence its preferred use in this study.

For the years 2016 to 2020, a total of 176 Sentinel-2 Level-1C (orthorectified top-of-atmosphere reflectance) images with sub-pixel multispectral registration (Baillarin et al., 2012) were acquired over HG (Fig. 1b; Table S1) obtained from ESA's Scientific Data Hub (https://scihub.copernicus.eu/dhus/#/home). These images covered the entirety of the study area on forty-four days between the months of May and September across the study period, equating to 1-2 images per month, allowing us to gain a full melt season perspective of supraglacial drainage evolution for the HG drainage catchment. For scenes with cloud cover below a 20% threshold, cloud coverage was typically restricted to the Kane Basin waterway (Fig. 1c) or ice interior locations beyond the melt extent, so did not pose any significant problems for river and lake mapping.

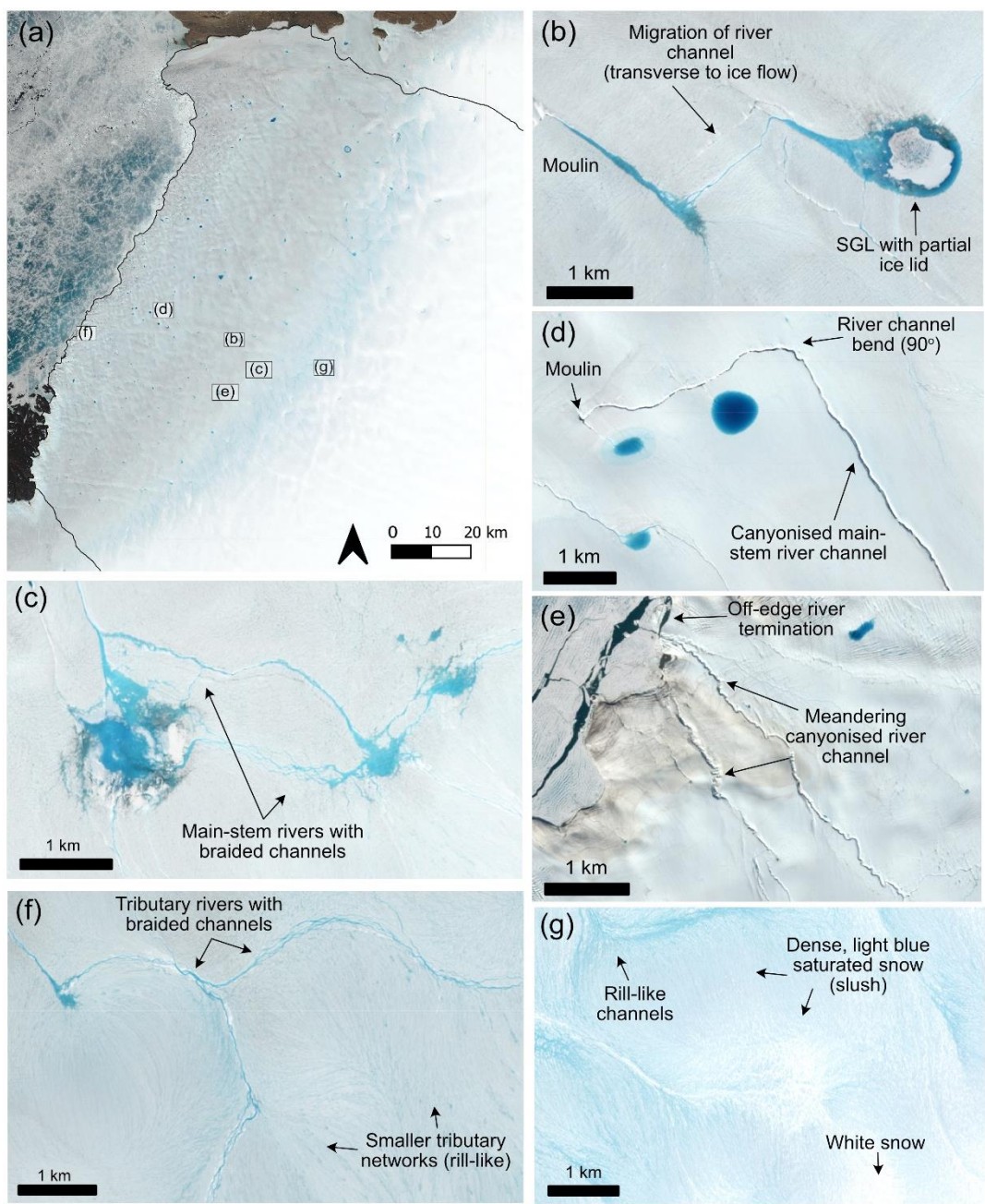

**Figure 2. Example of the supraglacial drainage features found across the study region of HG, represented in panel (a), from a Sentinel-2 image taken on 23rd June 2020 (RGB). (b) a supraglacial lake with a central ice-lid feeding an outlet supraglacial river, with evidence of river advection transverse to ice flow; (c) braided supraglacial rivers flowing between supraglacial lakes, known as 'connector' lakes; (d) a large canyonised supraglacial river with a 90-degree bend terminating abruptly in a moulin; (e) deep, canyonised supraglacial rivers flowing off the ice edge; (f) narrow supraglacial rivers with rill-features seen in the higher elevation regions of HG, flowing and coalescing into braided tributary rivers; (g) the slush zone, with dense areas of saturated snow (light blue) culminating in rill-like channels. The Sentinel-2 image is courtesy of the Copernicus Open Access Hub (https://scihub.copernicus.eu).**

The HG drainage catchment was generated using the ArcticDEM 10 m mosaic product obtained from the Polar Geospatial
Centre (https://www.pgc.umn.edu/data/arcticdem/) and delineated following the method of Karlstrom and Yang (2016).
Elevation contours at 100 m intervals were defined across the HG basin up to 1500 m a.s.l (maximum limit of the melt-prone
zone), equating to a size of 13,488 km$^2$. Daily surface meltwater production and runoff for the study area were extracted from
the Modèle Atmosphérique Régional (MAR) regional climate model (RCM) v3.11 (available at ftp://ftp.climato.be/; Fettweis
et al., 2017; 2020). MAR is among the best RCM to cover the GrIS as it explicitly models important polar processes (i.e.,
SMB) forced with ERA5 reanalysis data and has been extensively evaluated against in-situ automatic weather station and
satellite data (for a detailed description of MAR v3.11, see Amory et al. 2021; Fettweis et al., 2020). MAR has now been
widely used in other GrIS-based supraglacial hydrologic studies (Smith et al., 2017; Yang et al., 2019; Lu et al., 2020) for
quantifying the relationship between modelled-runoff and satellite-derived meltwater metrics for RCM accuracy validation.
The version MAR 3.11 used in this study was run at high spatial (6 km) and temporal (daily) resolution  to generate estimates
of daily meltwater production and runoff (R) in mm water equivalent per day (mm w.e. day$^{-1}$) within each grid cell to assess
the spatial and temporal distribution of meltwater against mapped supraglacial rivers and lakes, similar to other studies (Yang
et al., 2019; Lu et al., 2020; 2021; Yang et al., 2021). MAR grid cells were also sampled at each 100 m interval to assess
elevational gradients in both mapped drainage and runoff. A MAR uncertainty value of +/- 15% was also calculated (Fettweis
et al., 2020).
**3.2  Supraglacial river and lake extraction**
To effectively delineate supraglacial rivers from remotely sensed imagery, an automatic linear enhancement method developed
by Yang et al. (2015) was used, which characterises supraglacial rivers according to their Gaussian-like brightness cross
sections and longitudinal open channel morphology. Firstly, a Normalised Difference Water Index was performed following
McFeeters (1996) to differentiate active surface meltwater from the background ice and snow (Lu et al., 2020; Li et al., 2022).
This equation (Eq.1) utilises Band 3 ('Green') and Band 8 ('NIR') from Sentinel-2 imagery, as follows:
$$NDWI = \frac{(Green - NIR)}{(Green + NIR)} \qquad\qquad (1)$$
An ice-derived spectral index, $NDWI_{ice}$, has been widely applied to supraglacial mapping studies in recent years (Yang and
Smith, 2013; Moussavi et al., 2016; Williamson et al., 2018; Yang et al., 2021), as it produces fewer false classifications of
blue ice and slush areas. We use NDWI (Mcfeeters, 1996) here because of its successful implementation in other studies
(Stokes et al., 2019; Lu et al., 2020; 2021; Corr et al., 2022) and its ability to map all active surface melt of interest, including
slush zones. Additionally, preliminary testing of a small sample area found using NDWI (McFeeters, 1996) was able to map
16.3% more regularly connected supraglacial river channels compared to those mapped by the alternate $NDWI_{ice}$ (Fig. S1).
Next, an ice mask was applied, created from manually digitising the HG terminus from the latest, end-of-season image (late-
August/early-September) in each of the study years, to extract ice-only regions and remove surrounding land, rocky outcrops
and the ocean of the Kane Basin. A separate crevasse mask was delineated from manually identifying the heavily crevassed
zone known to extend up to 25 km from the northern sector of the terminus, with particular prevalence across the 7 km floating
section (Carr et al., 2015). This mask was applied and this section of the terminus removed to avoid the erroneous delineation
of crevasses and crevasse shadows in this area during image processing, as well as to reduce the effects of drainage
overestimation in these ice marginal regions in further calculations (Ignéczi et al., 2018). A global NDWI threshold of 0.4 ($t_{0.4}$)
was then applied to the masked NDWI image  as it effectively captures SGL boundaries and wide, main-stem river segments
(Lu et al., 2021). To aid in delineating narrower river segments and obtain a complete and continual supraglacial network, the
automatic river detection algorithm for linear enhancement (Yang et al., 2015) was then applied. This involved the removal of
the low-frequency image background and high-frequency image noise using a band-pass filter ramped between $1/200$ m$^{-1}$ and
$1/40$ m$^{-1}$ (Yang et al., 2019a), before the application of a Gabor filter to enhance the cross sections of smaller rivers (<2 pixel
width). This was followed by a parsimonious path opening (PPO) operator, which is a flexible mathematical morphological
operator, to stabilise linear brightness across river lengths and preserve connectivity, with a minimum path length of 20 pixels.

After Gabor-PPO filtering, the supraglacial river network becomes easier to differentiate and delineate from the surrounding
icy background. A global pixel brightness threshold of 5 (out of 255) denoted $t^5$, was used to extract supraglacial rivers of
varying widths from Gabor-PPO opened filtered images (Lu et al., 2020), i.e., from tributary-style rivers to main-stem, large
river channels. Hydrologically-connected  slush zones, which is where the pore space of snow becomes entirely water-saturated
when temperatures permit melting (above freezing, 0°C) and form expansive fields of ponded surface water, were retained as
they play an important role in the initial mobilisation of melt as slush flows or within rill-type channels and the inland expansion
of the melt-prone zone as summer progresses (Holmes, 1955; Marston, 1983; Cuffey and Paterson, 2010; Chu, 2014; Rippin
and Rawlins, 2021). Slush zones can be spectrally-distinguished in true colour satellite images as dense, light blue patches on
the surface as snow becomes water saturated (Fig. 2g; Holmes, 1955), and partially distinguished in NDWI images as bright,
dense features similar to that of individual linear river channels (Lu et al., 2021). For mapping conducted in this study, slush
zones are mapped inclusively within the drainage network and not treated independently due to their overall hydrologic
importance and spectral similarity to other hydrological components. Whilst dynamic thresholding techniques have been used
in other studies, particularly in the identification and mapping of supraglacial lakes across independent dates and/or years
(Selmes et al., 2011; Williamson et al., 2017), applying these two separate thresholds ($t_{0.4}$ and $t^5$) to all images across this study
period is reasonable for exploring the seasonal behaviour over the available dates used. Finally, these masks were vectorised
into separate river channel polylines and lake polygons for analysis, with a size threshold of 0.1 km$^2$ applied to SGLs. To assess
the ability of the automated mapping technique to capture supraglacial rivers at HG, a small area was manually digitised (Fig.
3). The overall spatial pattern extracted was similar and the difference between manual vs automated river area was 5.4%, with
automated rivers tending to have several gaps in the network compared to manual results. A similar finding was also produced
by Lu et al. (2020) who found a difference of 13.6% between manual vs automated supraglacial rivers also using Sentinel-2
imagery.

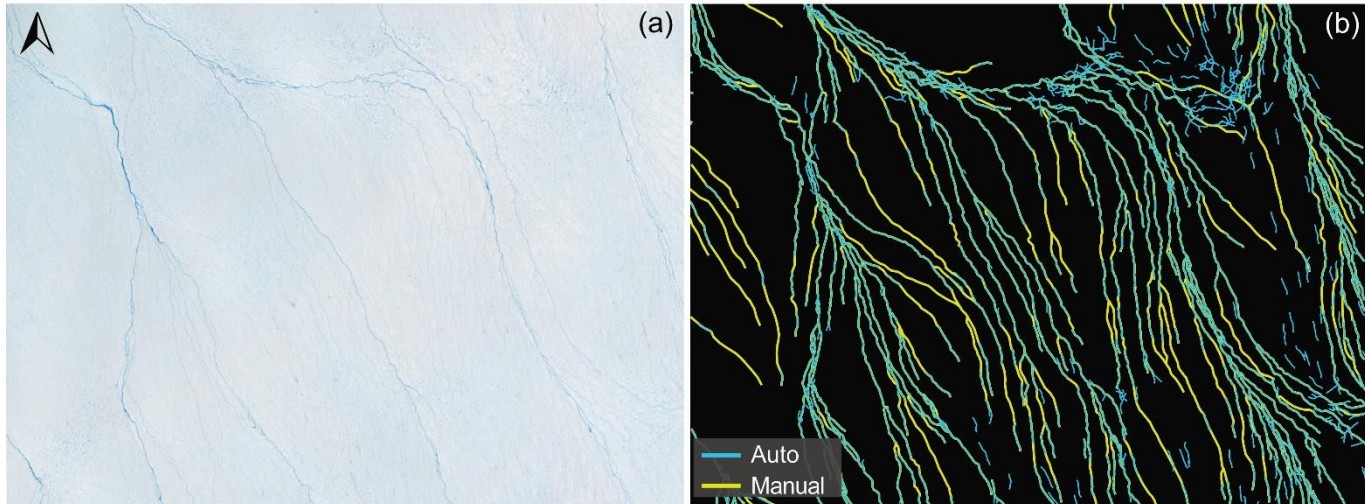

**Figure 3**. **(a) RGB image of a sample area of HG with both main-stem and tributary supraglacial river channels present; (b) a performance accuracy assessment comparing the automatic river detection algorithm (blue) used in this study with manually digitised networks (yellow). Overall, automated rivers networks were 5.4% shorter than those manually derived.**

## 3.3 Supraglacial river and lake quantification

To characterise the extracted supraglacial drainage system, metrics were calculated for both rivers and lakes. These metrics
are summarised in Table S2 and include meltwater area ($km^2$), meltwater area fraction (MF, %), drainage density ($D_d$), the
number of supraglacial lakes ($L_n$) and supraglacial lake area ($L_a$). MF is defined as the percentage total meltwater area across
the drainage catchment below a conservative upper melt limit of 1500 m a.s.l for each date mapped, which is then also further
divided into separate feature ratios including river area fraction (RF) and lake area fraction (LF). MF was also calculated across
100 m elevation contours from above the heavily crevassed zone at 200 m a.s.l, to the maximum melt extent at 1500 m a.s.l.
To explore the relationship between remotely-mapped drainage and modelled RCM MAR runoff (R), a Spearman's rank
correlation ($r_s$) was performed and linear regression analysis undertaken with subsequent $R^2$, $r_s$ and *P*-values reported.

## 4 Results

### 4.1 Spatial characteristics of the supraglacial drainage network

Supraglacial rivers and lakes were mapped from a total of forty-four dates across the lower, melt-prone 13,488 km$^2$ HG drainage basin from the melt seasons of 2016 to 2020. The mapped supraglacial drainage network across HG is shown extend up to 1440 m a.s.l, with well-developed, main-stem river channels occurring up to 1000 m a.s.l, which we characterise as the persistent zone, and an ephemeral network of tributary-style rivers and slush zones extending beyond 1000 m a.s.l in a transient zone up to maximum extent (Fig. 4a). Active supraglacial rivers and lakes form progressively up-glacier from low elevations (200 m a.s.l) to a maximum of 1440 m a.s.l as the melt seasons progress across the study period, with interannual variability observed.

Within the lower elevation regions (<400 m a.s.l) of HG, collectively the supraglacial drainage network is largely fragmented, with many short (<750 m long), supraglacial rivers observed alongside small SGLs with an average size of 0.23 km$^2$: the smallest observed across all elevation bands. At greater elevations, beyond 400 m a.s.l, we observe large, main-stem supraglacial rivers, some with incised-canyon features (Fig. 2b), interconnected with increasingly larger SGLs. At elevations >1000 m a.s.l, average SGL size is 0.41 km$^2$, with a maximum SGL size of 2.08 km$^2$. SGLs and rivers parallel to ice flow tend to be highly persistent year-on-year across the study period. In Figure 4b, we also see some evidence of a potential main-river reconfigurations, with the north-westward advection of a river channel that runs transverse to ice flow. In the upper parts of the catchment, the supraglacial drainage network becomes increasingly dense, especially between 800 – 1000 m a.s.l, where we see a 120% increase in average meltwater area (94.5 km$^2$) compared to 200 – 400 m a.s.l (42.9 km$^2$), a 79% increase compared to 400 – 600 m a.s.l (54.2 km$^2$) and a 10% increase compared to 600 – 800 m a.s.l. Not only are persistent main-stem rivers still present up to 1000 m a.s.l, but an extensively connected tributary river system is also observed within this persistent zone.

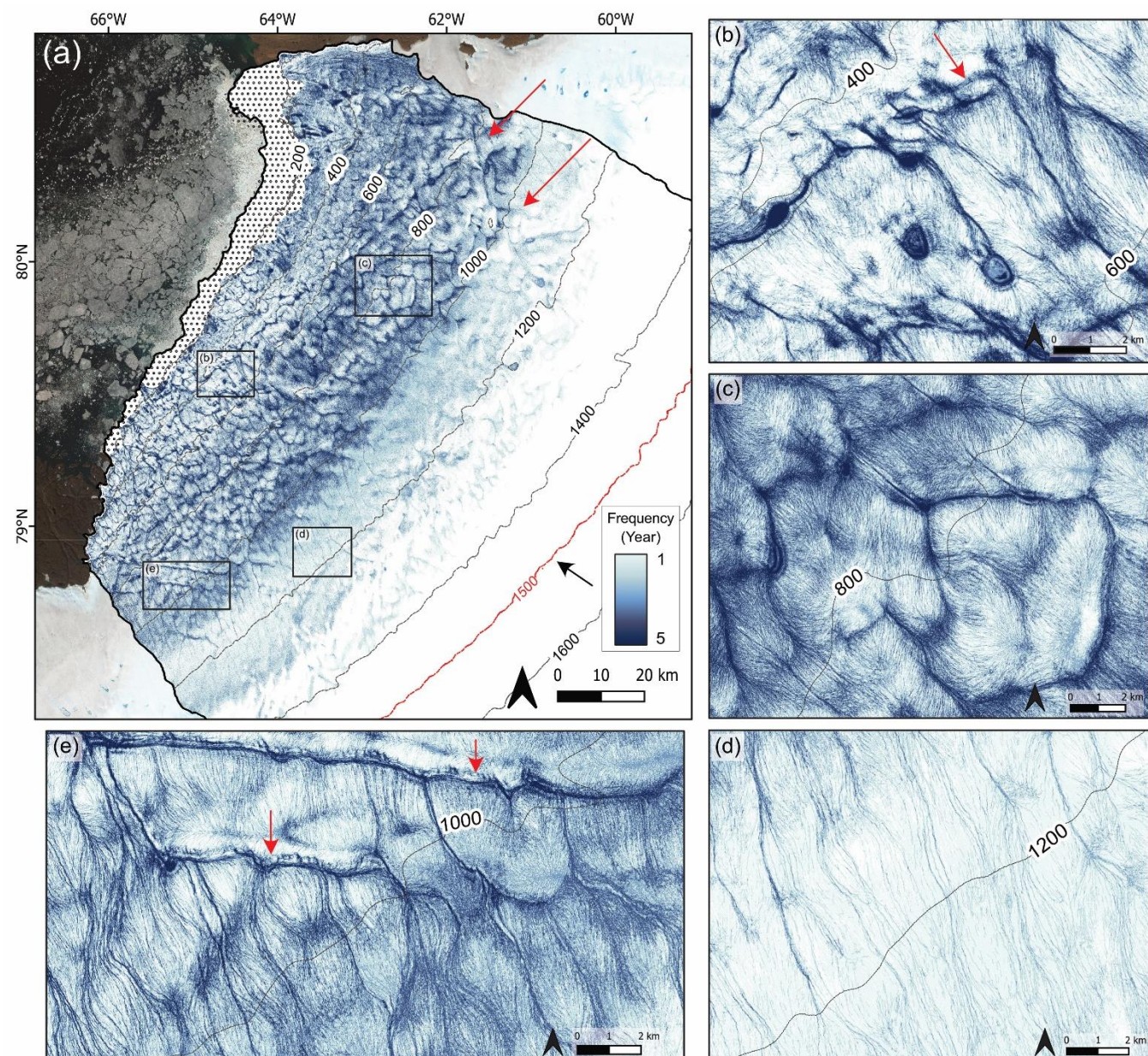

**Figure 4. – (a) Map showing the recurrence frequency of the supraglacial drainage system across the study period (2016 to 2020) at HG. Dark blue shades denote a higher frequency of occurrence, which is highly prominent in the persistent zone below 1000 m a.s.l, with rivers and lakes typically reforming in the same location each year. Red arrows denote the location and direction of two distinct parallel drainage structures. Black arrow represents ice flow direction. The background Sentinel-2 image courtesy of the Copernicus Open Access Hub (https://scihub.copernicus.eu); (b) close-up panel showing highly persistent rivers in the persistent zone with a prominent 90-degree angle in their channel form. Red arrow denotes some channel reconfiguration due to ice advection; (c) close-up panel showing the dendritic supraglacial drainage structure typical in the northern sector of HG; (d) close-up panel of the transient zone (> 1000 m a.s.l) where yearly river persistence is lower and characterised by lengthy tributary river channels; (e) close-up panel showing the more parallel-form of supraglacial drainage structure in the southern sector of HG with red arrows denoting some channel advection with ice motion.**


The drainage network up to 1000 m a.s.l, in particular within the central and northern reaches of HG, is dendritic in nature
(Fig. 4c). This type of drainage, however, is not uniform across HG, with the supraglacial network to the south exhibiting a
more sub-parallel drainage style (Fig. 4e), with this configuration extending beyond the persistent zone and into the more
transient zone with slush zone development and inland evolution a key part of the temporal aspect of the network at higher
elevations (Fig. 4a). Even though density remains high until 1300 m a.s.l, vastly transient tributary rivers and slush zones
dominate, feeding meltwater from headwater regions downstream. In terms of SGLs, the maximum recorded elevation was
1346 m a.s.l both in 2019 and 2020. Fewer SGLs are observed in the transient zone, accounting for 16% of total SGL area
across the study period. However, average SGL area increases with elevation, with lakes in this zone being 54% larger (0.4
km$^2$) on average than those found below 1000 m a.s.l (0.26 km$^2$). The single largest SGL size of 2.08 km$^2$ was recorded in
2016 at 1150 m a.s.l.

A key feature that is particularly prominent in the supraglacial drainage network is the presence of two parallel lines that track
across-glacier from a south-west to north-east direction (Fig 4a). Many supraglacial rivers and lakes are aligned along these
two features which appear as depressions in the ice surface, with the abrupt termination of many rivers indicating meltwater
capture via moulins, indicating a strong structural element influencing drainage configuration.
**4.2 Temporal evolution of the supraglacial drainage network**
Typically, the supraglacial drainage network becomes active in early-June with the on-set of melt production and runoff in the
region, with only a small number of large-stem supraglacial rivers becoming active and subsequently recorded (MF <3.2%)
during this time within the mapped elevation bands of 200 – 950 m a.s.l. By late June a widespread (500 m to 1150 m a.s.l)
slush zone develops and advances up-glacier as the melt season progresses runoff increases, with MF ranging between 11.4
and 19.9%. As bare ice is exposed below this slush zone, the drainage system becomes increasingly channelised (Fig. 5). The
formation of the slush zone at the end of June typically coincides with maximum melt storage in SGLs (both numbers and size
of lakes). The network with the largest expanse of slush zone and number of SGLs in this study was observed on 30[th] June
2019 (Fig. 5), with 2685 km$^2$ (19.9%) of the HG ice surface comprised of a hydrologically-connected, unchannelised system
and 111 SGLs recorded (total area 27.4 km$^2$). As the season progresses, the slush zone shifts upglacier whilst reducing in size,
with average June MF decreasing by up to 33% before stagnating at a maximum inland extent, ranging between 1050 m a.s.l
(2018) and 1440 m a.s.l (2020) across the study period. At this elevation, the slush zone operates thereafter as headwaters,
feeding the complex, transient, tributary systems below and further supling the larger, well-defined supraglacial rivers at
lower elevations (<1000 m a.s.l).

Towards the end of the melt season, despite melt and runoff cessation, the supraglacial drainage network remains (Fig. 5).
The interannual variability in seasonal behaviour of the supraglacial drainage network between 2016 and 2020 (Fig. 5)
corresponds to the length and intensity of the melt season. Drainage within the melt seasons of 2016, 2019 and 2020 follow a
similar pattern characterised by a rapid increase and peak in MF in late-June, yielding values of 11.4% (R = 7.1 mm day$^{-1}$),
19.9% (R = 19.4 mm day$^{-1}$) and 12.1% (R = 19.6 mm day$^{-1}$) respectively (Fig. 6); concurrent with early-season melt production
and runoff. These high MF values are largely associated with widespread slush zone initiation, with a subsequent peak in MF
increasing the drainage network area by 267% in 2016 (28[th] June) and 322% in 2019 (30[th] June), with its spatial extent observed
below 1000 m a.s.l at this time (Fig. S3). The beginning of the melt season in 2020 is an exception, with an anomalously high
MF (11.6%) recorded on the 15[th] June reaching 1100 m a.s.l, as well as a high number (57) and cumulative area of SGLs (11.2
km$^2$; Fig. 6). This high is followed by a 66% reduction in MF by the 23[rd] June, after which the networks behaviour is similar
to that of other seasons, with a subsequent MF increase of 211% to its peak at the end of June (28[th] June; Fig. 6). Despite high
rates of melt production and runoff throughout July across these three melt seasons, MF plateaus and reduces, reaching a
steady-state of between 7.6% and 13.9%. The number of SGLs also reduces on average between 20% and 27% throughout
July, with cumulative SGL area reducing between 9% and 38%.

By the end of July, the supraglacial drainage network consistently extends to 75 - 80 km inland (1440 m a.s.l) at its maximum
areal extent.  As melt and runoff reaches declines into August, the drainage network reduces between 53% (2016) and 9%
(2020). In 2019, persistent high rates of melt production and runoff result in persistence of the drainage networks maximum
inland extent and even a late-season increase of 54% in MF from the 5[th] August to 13[th] August. 2019 is registered as an
exceptional year at HG, with an average meltwater area up to 75% greater in June compared to 2016, between 25% and 92%
greater in July compared to 2016 and 2020 and up to three times greater (300%) in August compared to both 2016 and 2020.
Additionally, 129% and 86% more lakes are recorded compared to 2016 and 2020 respectively, with SGLs persisting much
later into the melt season (mid-August) than the other years.

The supraglacial drainage network in 2017 and 2018 behaves quite differently to the other melt years (Fig. S4). Until late-July,
MF remains low (<7%; Fig. 6) with runoff predominately occurring within the lower 700 m of the HG basin (Fig. S3). The
number and area of SGLs is also low during this period, with an average of 15 lakes observed in June and July across 2017
and 2018 with a cumulative average area of 21.5 km$^2$ and 31.1 km$^2$ respectively; 280% lower than average lake counts and
218% lower than cumulative average area for the same monthly periods across 2016, 2019 and 2020. Unlike the other study
years, peak MF does not occur until much later in the melt season, reaching 10.2% on 22$^{nd}$ August 2017 and 7.2% on 20$^{th}$
August 2018 with a 690 km$^2$ (197%) and 521 km$^2$ (135%) increase in meltwater area to its prior mapped date respectively;
concurrent with peak runoff and the widespread occurrence of a late slush zone.  Similarly, SGL numbers and area also peak
on these dates, with 54 (15.8 km$^2$) and 74 (20 km$^2$) lakes recorded, comprising 32% and 39% of total lakes recorded per year
respectively. Examination of the maximum extent of the supraglacial drainage network is also revealed to be limited, with a
maximum elevation of ~1150 m in 2017 and ~1050 m in 2018, equivalent to between 48 and 51 km inland.

When comparing what could be considered as high vs low melt year seasonal patterns, the supraglacial drainage network is
twice as large during high melt years (2016, 2019 and 2020) and is seen to form at elevations ~300 m higher than in low melt
years (2017, 2018; Fig. S2 and S3). In terms of the average number of SGLs, there is almost double the number of SGLs
observed (92%) in the high melt years compared to low melt years with average SGL area also being 111% higher, showing
large year-on-year variability of this system.

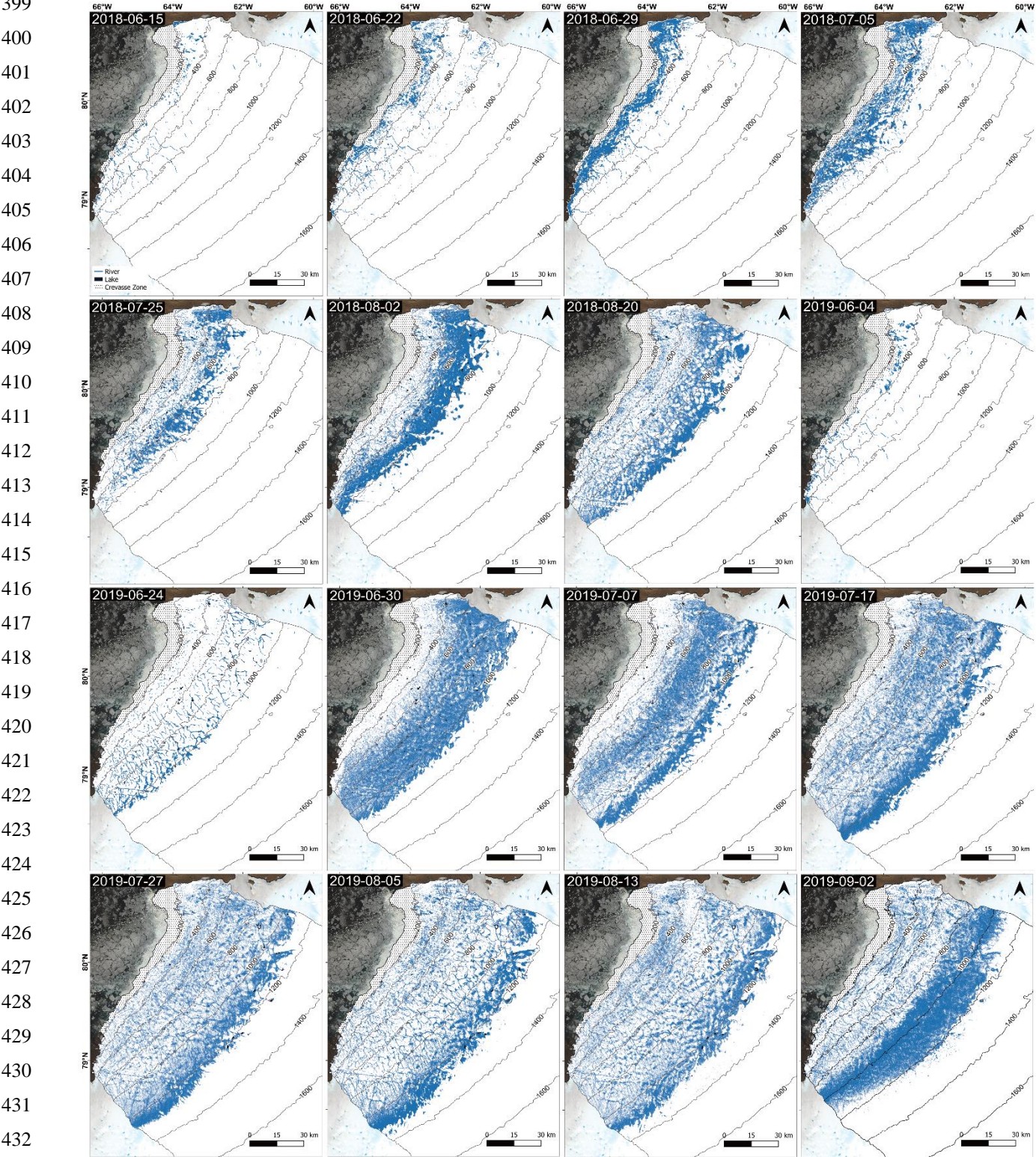



**Figure 5. Satellite-derived mapping of the temporal evolution of the supraglacial drainage network, including rivers and lakes, across the HG drainage basin from two melt years during the study period; 2018 showing the typical behaviour of a low melt year and 2019 showing the behaviour during a high melt year. The background image is a Sentinel-2 image courtesy of the Copernicus Open Access Hub ([https://scihub.copernicus.eu](https://scihub.copernicus.eu)). For all mapped study dates, please refer to Fig. S3.**



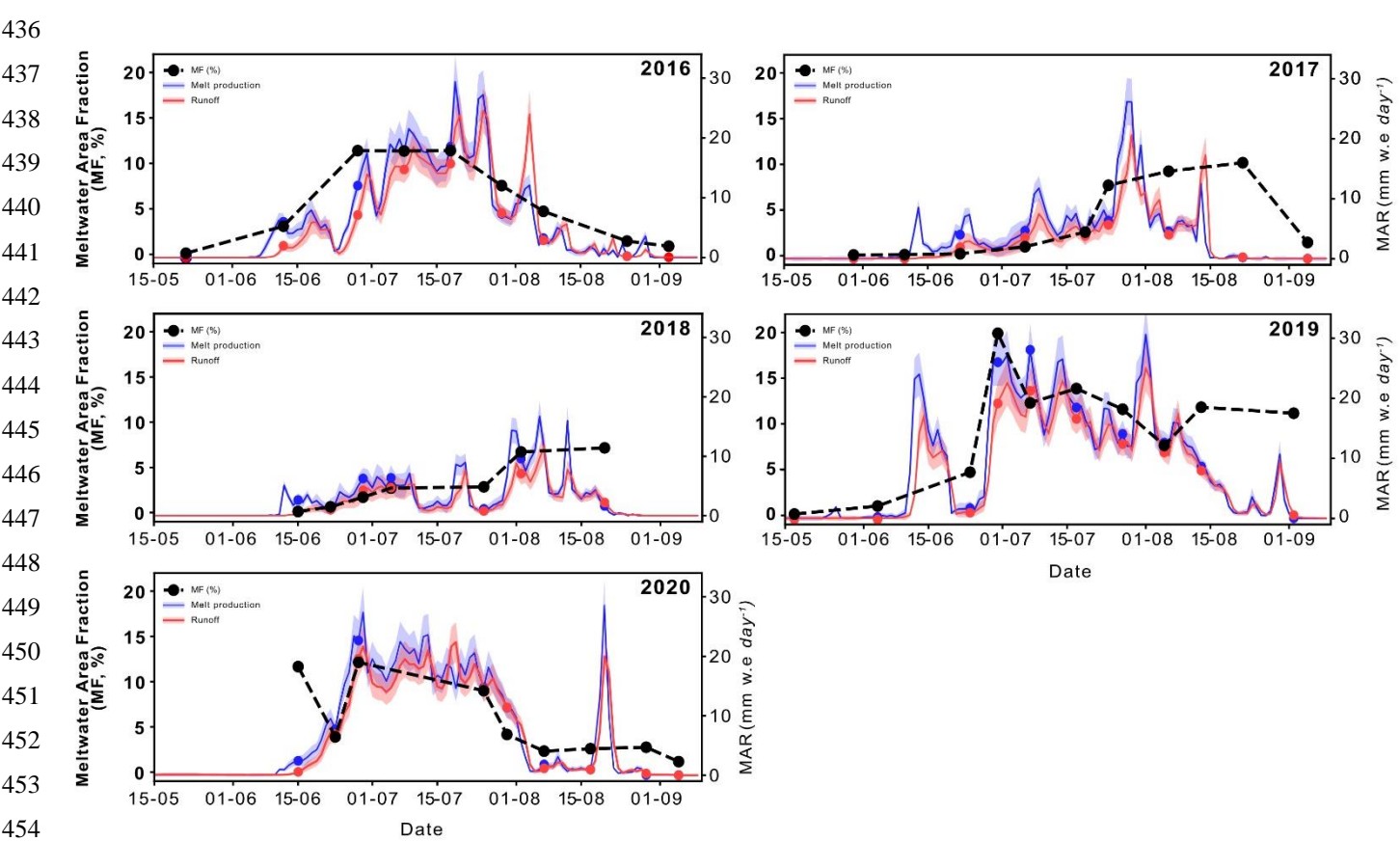




















**Figure 6. The satellite-derived water metric meltwater area fraction (MF, %) for each mapped date across the study period (2016-2020) alongside MAR v3.11 derived melt production and runoff values (mm w.e day$^{-1}$) for the HG catchment. +/- 15% uncertainty envelopes are provided for MAR-derived values (Fettweis et al., 2020).**


The contribution of rivers and SGLs to the supraglacial drainage network follows the same general seasonal trend as MF. RF
and LF (%) peak at the end of June in the high melt years of 2016, 2019 and 2020 and in mid-August for the low melt years
of 2017 and 2018 (Fig. 7): when hydrologically-connected slush zones are most prominent and SGL numbers are shown to
peak. Overall, the network is largely composed of supraglacial rivers, with RF accounting for an average of 6.2% ± 5.6%
(mean ± std) of the HG catchment across the study dates and is higher during high melt years (7.8% ± 6.0%) compared to low
melt years (3.5% ± 3.4%). The highest RF recorded occurred on the 7[th] July 2019 and measured 24.5%; the highest RF
measured in a supraglacial hydrologic study. For SGLs, average LF was much lower, accounting for an average of 0.07% ±
0.06% of the melt-prone area of the catchment, with slightly higher LF in high melt years (0.08% ± 0.07%) compared to low
melt years (0.05% ± 0.04%). In terms of the overall contribution of supraglacial rivers and SGLs to peak meltwater area
fraction, 98% of the network is dominated by supraglacial rivers across all study years, with SGLs playing a less dominant
role in HG's drainage network. In comparison, rivers contribute ~62% to the drainage network in southwest Greenland, with
SGLs contributing ~38% (Yang et al., 2021).4.3 Supraglacial hydrology and MAR runoff

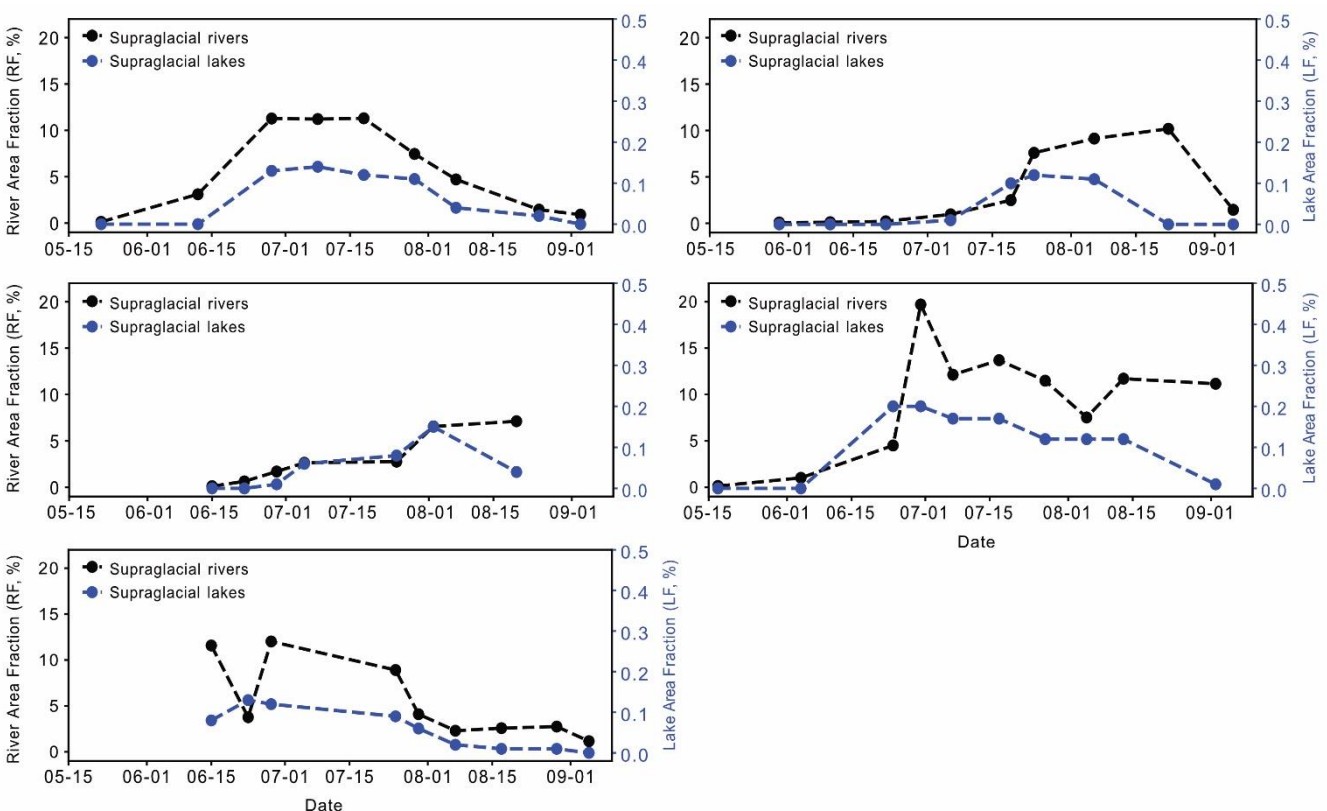

**Figure 7. River area fraction (RF, %) and lake area fraction (LF, %) across the melt-prone zone of HG for each mapped date across the study period. On average, 98% of the supraglacial drainage system across the study period at HG is comprised of RF.**

### 4.3 Supraglacial hydrology and MAR runoff

A strong positive linear relationship was identified between satellite-derived MF and regional climate model MAR surface R for the HG catchment across the study period 2016 to 2020 (Fig. 8: $R^2 = 0.77$, $R_s = 0.91$, p = <0.001) up until peak MF values and rapid surface runoff decline at the end of the melt season. Both MF and R increased concurrently each year as the melt season progressed, with peak runoff often coinciding with the existence of expansive slush fields across the upper part of the catchment. Runoff remained high until maximum extent occurred, particularly for high melt years: 2016 (29th July, R = 7.5 mm day$^{-1}$), 2019 (13th August, R = 8.2 mm day$^{-1}$) and 2020 (30th July, R = 11.4 mm day$^{-1}$). For low melt years, runoff remained relatively high until early August (6th August 2017, R = 4.0 mm day$^{-1}$; 2nd August 2018, R = 7.1 mm day$^{-1}$), with maximum extent occurring within two-weeks (the next mapped date). This relationship between MF and R shows the reliability of simulated variations in seasonal surface meltwater runoff in capturing the behaviour of the supraglacial drainage network via satellite-derived water metrics, particularly during high melt years and until runoff declines each melt season.

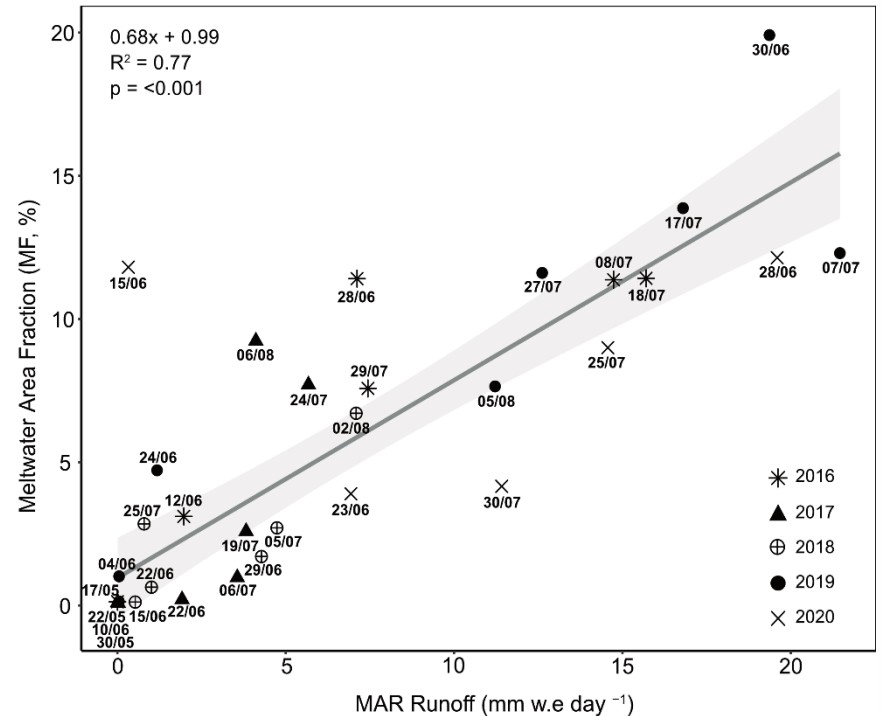

**Figure 8. Linear relationship between the satellite-derived water metric meltwater area fraction (MF, %) and RCM-derived runoff from MAR v3.11 for each year studied up until runoff declines.**

## 5 Discussion

### 5.1 Spatial characteristics of the supraglacial drainage network

Our satellite-derived mapping reveals a complex supraglacial drainage network at HG that reaches a maximum elevation of 1440 m a.s.l, recorded in 2020. This maximum elevation is 400 m lower than supraglacial drainage observations in southwestern Greenland (~1800 m a.s.l; Yang et al., 2021), consistent with observations by Ryan et al. (2019) who found the maximum snowline elevation to be 400 m lower in northern Greenland than southwestern Greenland. Up to elevations of 1000 m a.s.l, the supraglacial drainage network is highly persistent across the study period, with a variable system beyond this elevation observed within the transient zone; defined in this study as a high elevation region where drainage is transient in time, but not necessarily space. The recurrence frequency of meltwater pixels in the supraglacial drainage network (Fig. 4) demonstrates the stability of SGLs in particular, which re-occupy the same location year-on-year despite ice advection (Smith et al., 2015; Pitcher and Smith, 2019). SGLs increase in size with elevation, with higher elevation lakes persisting for longer (i.e., incapable of draining) and undergoing greater season expansion (Gledhill and Williamson, 2018; Yang et al., 2021). Larger lakes at higher elevations are the result of longer wavelength bed undulations being favourably transferred to the ice surface through thicker ice creating large, shallow surface depressions (Ng et al., 2018). Unlike SGLs found <400 m a.s.l, which are smaller and have likely reached their maximum available melt area, higher-elevation lakes are not yet topographically constrained (Krawczynski et al., 2009). This is similar to observations in southwestern Greenland where SGL size was larger and more variable >1400 m a.s.l, with 21% of these lakes draining via hydrofracture (Yang et al., 2021).

Whilst SGL location is known to be largely controlled by bed topography (Lampkin and Vanderberg, 2011; Igneczi et al., 2018), this study also notes that many well-established rivers that are longitudinal to ice flow, including many with canyonised features, also reoccupy locations. Supraglacial rivers that are transverse to ice flow or have a transverse element to their channel, however, may be less stable in some areas by up ~300 m (Fig. 4a) over the study period, probably due to the impact of ice advection. In the transient zone, the recurrence frequency is reduced, with tributary rivers and slush zones dominating at higher elevations. Here, their persistence is highly reliant on there being enough melt at higher elevations to initiate and sustain channel formation, which in 2017 and 2018, was limited and therefore drainage occurrence was much reduced in this zone. Also, tributary rivers are typically lower order with narrower channels and shallower depths (Smith et al., 2015; Pitcher and Smith, 2019), meaning their form has the potential to migrate, close and reform quickly, if melt is available. The transient zone is therefore not only influenced by melt availability overtime, but the potential for migration in space. In agreement with previous studies (Joughin et al., 2013; Poinar et al., 2015), we also show that rivers tend to be longer at higher elevations (>25 km long), consistent with observations in SW GrIS (>40 km long), likely due to the basal transfer of only long-wavelength basal undulations to the surface due to thicker ice and the reduced presence of surface crevassing (Gudmundsson, 2003; Lampkin and Vanderberg, 2011; Crozier et al., 2018).


Drainage patterns are also shown to vary across HG, with a dendritic-style of drainage observed in the northern sector and a
parallel-style drainage observed in the southern sector. These differing patterns not only highlight variations across different
hydrologic catchments of the GrIS, but also intra-catchment variations, which may stem from local variations in surface
topography via the transmission of basal topography (Raymond and Gudmundsson, 2005; Ng et al., 2018) controlled by bed
roughness/ structure, wavelength transfer and differing ice flow regimes (Gudmundsson et al., 1998,2003; Lampkin and
VanderBerg, 2011; Crozier et al., 2018; Igneczi et al., 2018). This has been shown to play an important role in mapped rivers
and lake hydromorphology at both 79°N Glacier (Lu et al., 2021) and across the Devon Ice Cap (Wyatt and Sharp, 2015).

In the northern sector of HG, we observed short supraglacial rivers and small SGLs at lower elevations (200 - 400 m a.s.l) and
a prominent dendritic-style drainage pattern of interconnected rivers and lakes up to 1000 m a.s.l, with some larger rivers
abruptly terminating. Such characteristics suggest the interception of runoff by crevasses and moulins, with such capture
directed to the en- and sub-glacial system with the potential for pronounced impacts on localised flow rates (Catania et al.,
2008; Schoof, 2010; Mejia et al., 2022). This drainage style is typical of that observed within the western and southwestern
sectors of GrIS, whereby structurally-controlled drainage of supraglacial rivers flowing between SGLs promote shorter river
channels, whilst high rates of crevassing at lower elevations (<1000 m a.s.l) means virtually all surface meltwater is captured
and diverted before reaching the ice edge. This more compact style of drainage is likely to create a 'flashier' response in
hydrographs, with a greater runoff peak and shorter rising limb (Smith et al., 2017). Additionally, as shown by Carr et al.
(2015) and Rignot et al. (2021), this sector of HG sits within a 475 m deep basal trough that extends ~45 km wide and >70
km inland and is characterised by fast rates of flow (200 – 600 m yr$^{-1}$). Faster basal sliding has the ability to promote the more
efficient transfer of basal topography to the surface and can subsequently precondition the large-scale spatial structure of the
surface drainage system, including drainage fragmentation due to higher rates of crevassing and moulin formation, which can
further supply meltwater to this faster-flowing sector (Crozier et al., 2018; Igneczi et al., 2018).

In comparison, the sub-parallel drainage structure of the supraglacial network in the southern sector differs greatly to that of
the northern sector of HG and observed drainage in western and southwestern regions of the GrIS. Drainage largely consists
of continuously-flowing rivers that drain surface meltwater from the slush zone at ~1500 m a.s.l to much lower elevations (200
m a.s.l), with some rivers directly terminating off the ice sheet edge. This suggests limited opportunities for meltwater to
penetrate to the ice sheet bed, with meltwater having longer transport times to travel to the catchment outlet (i.e., proglacial
zone/Kane Basin) with the hydrograph expected to have a more subdued and longer rising limb (Karamouz et al., 2013; Yang
et al., 2019a). Within this southern sector of HG, ice velocity is significantly slower (<100 m yr$^{-1}$; Rignot et al., 2021) than its
northern counterpart, with relatively thick ice contributing to the absence of crevasses and moulins (Oswald and Gogineni,
2011; Yang et al., 2019a; Andrews et al., 2022), as well as controlling the hydromorphology of the drainage network found
here. Similar drainage hydromorphology was also mapped at the neighbouring glacier at Inglefield Land (Yang et al., 2019a;
Li et al., 2022), with supraglacial rivers flowing uninterrupted into the proglacial zone, with discharge into terrestrial rivers
directly reflecting the timing and intensity of surface meltwater runoff from the outlet glacier catchment without modification
from en- and/or subglacial processes.

The supraglacial drainage configuration is also further influenced by significant structural elements which were identified by
Livingstone et al., (2017) via the Moderate-Resolution Imaging Spectroradiometer (MODIS) mosaic of Greenland (Haran et
al., 2013). Two linear structures expressed as depressions on the ice surface that run in a southwest to northeast direction across
HG are clearly visible on the mapped glacier surface in this study (Fig. 4a, Fig. 9), with many supraglacial rivers and SGLs
aligned-to or terminating at them. Some longitudinal rivers are also shown to suddenly change direction when encountering
these structures, with subsequent channels diverting at a 90-degree angle, transverse to ice flow (Fig. 4a, 4b). It is at the
intersection of such structures we observe some channel advection with spacing of 300 m (Fig. 4a), broadly representing the
ice displacement over the study period. Other basal structures are also reflected within the supraglacial drainage system,
including many V- and X- shaped patterns clearly controlled by depressions in the bed. There is a strong glacier-wide emphasis
here of these structures influential control on drainage, which are reproduced here in Figure 7 based on Livingstone et al.
(2017) from MODIS imagery (MOG2015, Haran et al., 2018) and also within bed topography data from BedMachine (version
4; Morlighem et al., 2017, 2021). This identification provides independent confirmation of the existence of these depressions
in the bed and their subsequent expression on the surface, as well as how they significantly control the multi-year surface
drainage configuration within the vicinity of such structures.





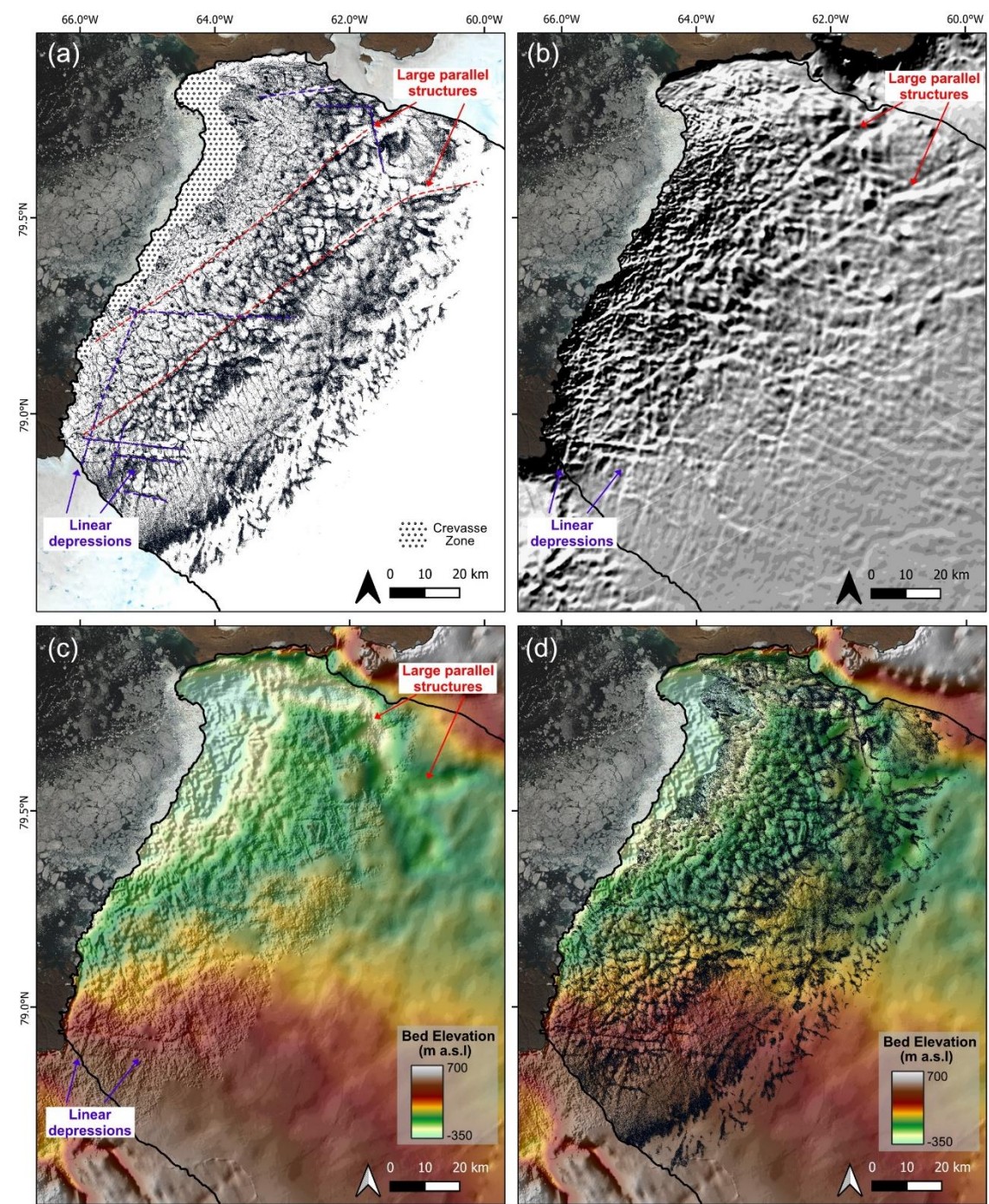

Figure 9. (a) Combined map of the supraglacial drainage network at its maximum extent across 2016 to 2020 showing the two parallel structures orientated southwest to northeast identified by red dashed lines. Other V- and X-shaped structures in the drainage network are also highlighted by purple dashed lines; (b) Moderate-Resolution Imaging Spectroradiometer (MODIS) Mosaic of Greenland (MOG2015, Haran et al., 2018) showing evidence of the these structures on the ice surface; (c) Bed topography via BedMachine (Morlighem et al., 2020) showing the structures as depressions within the bed; (d) the supraglacial drainage network as presented in (a) overlain on top of bed topography presented in (c) showing the overall influence bed topography has on the surface drainage structure at HG.

The supraglacial drainage network at HG is also shown to be largely dominated by supraglacial rivers (98%), consistent with findings by Lu et al. (2021) in northeast Greenland where supraglacial rivers also dominate the surface drainage network (83.8%). Such river contributions in northern Greenland are significantly higher than southwestern GrIS, whereby supraglacial rivers account for an average of 62% of the network (Yang et al. 2021). The difference in contribution of rivers and SGLs to the drainage network between north and southwest Greenland is likely the result of (i) reduced lake distributions and storage >800 m a.s.l in northern Greenland due to surface topographic constraints (i.e.,reduced number of available depressions due to ice thickness; Ignéczi et al., 2016; Lu et al., 2021), (ii) steeper northern surface slopes promoting meltwater runoff over storage, with gentler elevation gradients in southwest Greenland promoting a wider, more susceptible ablation zone to melt and meltwater ponding (Mikkelsen et al., 2016; van As et al., 2017). At HG, the maximum supraglacial RF recorded for a single date across the melt-prone zone reached 24.5%: the largest RF value recorded to-date from a supraglacial drainage mapping study (e.g., 5.3% in northeast Greenland, 5.5% at the Devon Ice Cap and ~14% for southwest Greenland). A reason for this large value is likely due to the inclusion of expansive hydrologically-connected slush zones within hydrologic mapping, which are prevalent during the early-to-mid melt season when snow becomes water saturated. This slush zone progressively moves upglacier and is removed to reveal the bare ice below. Additionally, a lower river threshold ($t^5$) was used in this study compared to other studies (e.g., $t^{20}$, Lu et al., 2021), allowing narrower supraglacial rivers and slush zones to be captured, increasing the overall supraglacial drainage and river components. Whilst supraglacial rivers are likely to dominate the drainage network here, similar to northeast Greenland (Lu et al., 2021) regardless of the threshold used due to reduced lake presence, care should be taken when directly comparing drainage network statistics between mapping studies from differing regions of the GrIS.

## 5.2 Temporal evolution of the supraglacial drainage network

Across the study period (2016 – 2020), we observe the seasonal development and inland evolution of the supraglacial drainage network as the melt season progresses and runoff increases up to the maximum melt extent, which typically occurs at the end-

of-July. During the early melt season and at higher elevations as melt progresses, we observe the growth and inland migration of a large, poorly channelised slush zone (Greuell and Knap, 2000). Within this zone, slush flows can occur as surface melt percolates and saturates the snowpack, promoting slush mobilisation into topographical lows and initiating the reopening of perennially-occupied channels (Cuffey and Paterson, 2010; Irvine-Fynn et al. 2011). On higher-draining slopes, such mobilisation can form slush-filled rills, which coalesce into networks of arborescent tributary channels, efficiently transporting melt to larger, primary river channels and, subsequently, the transportation of melt down-glacier (Marston, 1983; Cuffey and Paterson, 2010; Chu, 2014; Rippin and Rawlins, 2021). The seasonal development of the drainage network is shown to transform from a system initially dominated by water percolation to one dominated by channelised, efficient flow (Fig. 5); further confirming behaviour identified across multiple supraglacial drainage mapping studies across the GrIS (Lu et al., 2021; Yang et al., 2021; Li et al., 2022).

The rate and extent of the spatial and temporal evolution of the supraglacial drainage network is highly variable between years. Several years within the last decade have been characterised by high air temperatures and extreme melt events, including two years represented within this study; 2016 and 2019. Both years, in particular 2019, experienced a strong negative North Atlantic Oscillation and simultaneously a positive East Atlantic index and Greenland Blocking Index phase, which are associated with persistent, anticyclonic conditions over Greenland driving enhanced surface mass loss (Lim et al., 2016; Cullather et al., 2020; Zhang et al., 2022). Mass loss during summer 2019, in particular, was promoted by enhanced solar radiation, reduced cloud cover and the north-westward advection of warm, moist air from the western margins as a result of such atmospheric variability (Hanna et al., 2021; Cullather et al., 2020; Tedesco and Fettweis, 2020; Elmes, 2021). Combined with low snow accumulation in the 2018/19 winter (Sasgen et al., 2020), extensive melting occurred along much of the Greenland coast, with surface melt experienced in the north being the highest on the record since 1948 (Tedesco and Fettweis, 2020). It was during this exceptional and long melt year that we observed the highest MF values (19.9% or 2685 $km^2$ recorded on the $30^{th}$ June 2019) and second highest areal extent of supraglacial drainage network (1375 m a.s.l; Fig. 5). Ablation continued throughout September (Sasgen et al., 2020; Tedesco et al., 2019; Tedesco and Fettweis, 2020), however this was beyond our mapped timeframe. The year 2019 also recorded the largest number of SGLs (527) and cumulative lake areas (151.8 $km^2$). The hydrologic expansion of the drainage network was also rapid, in-line with a record early-melt season event, which combined with low snow accumulation (Tedesco and Fettweis, 2020), promoted rapid snowpack warming, disintegration and exposure of the bare ice zone, resulting in an enhanced melt-albedo feedback mechanism (Tedesco and Fettweis, 2020). Similar findings by Turton et al. (2021) and Hochreuther et al. (2021) at 79°N Glacier recorded the largest SGL numbers and extents of their study periods in 2019, indicating the widespread impact of this extreme melt event, particularly across the GrIS northern sector.

Similar seasonal and multi-annual behavioural patterns during 2017 and 2018 were, again, also observed by Turton et al. (2021) and Lu et al. (2021) on the north-eastern glacier of 79°N and Otto et al. (2022) on the northern Ryder Glacier. These findings all record a slow rate of SGL increase and late area peak (early-August) in 2017 and 2018, with Lu et al. (2021)

confirming the late area peak in combined supraglacial drainage mapping of both lakes and rivers during August 2017. SGL
mapping by Turton et al. (2021) also identified the delayed development and lower SGL presence during the 2018 melt season,
with SGLs largely limited to <900 m a.s.l. Such findings are consistent with observations in this study, with the onset and
inland evolution of the supraglacial network, including both rivers and lakes, delayed by ~1 month compared to other the other
study years (2016, 2019, 2020) and the limited areal development of the network (<1150 m a.s.l). Both the melt seasons of
2017 and 2018 recorded below average melt (1981 – 2010 reference period) and melt extents (32.9% and 44% respectively)
across the GrIS (Tedesco et al., 2017, 2018; Sasgen et al., 2020). There was heavy springtime snowfall and late surviving snow
in bare ice areas (Tedesco et al., 2017, 2018); consistent with a strongly positive average summer (JJA) North Atlantic
Oscillation and a negative Greenland Blocking Index, hypothesised to inhibit surface melt and promote increased summertime
snowfall (Ruan et al., 2019; Sasgen et al., 2020), hence these anomalously cold summers.

The anomalously early spike in satellite-derived meltwater area recorded in early-June 2020 raises questions as to how extreme
melt years, such as 2019, may precondition the ice surface (Cullather et al., 2020) and affect surface conditions and subsequent
surface hydrologic behaviour the following year (Culberg et al., 2021). Some SGLs on the GrIS have been found to persist
throughout the winter months, due to insulation from a layer of ice and/or snow (Koenig et al., 2015; Law et al., 2020; Schröder
et al., 2020). This lake persistence includes the winter of 2019/2020, when late-summer surface melt and high autumnal
temperatures (August – November) are believed to have increased subsurface firn temperatures, delaying and even decreasing
the ability for subsurface meltwater freezing in northern Greenland, contributing to higher totals of liquid-buried SGLs
(Dunmire et al., 2020). This alongside a drier-than-average (1981 – 2020) winter and spring (Tedesco and Fettweis, 2020;
Moon et al., 2020), has the potential to cause the rapid disintegration of limited snow present and the subsequent swift exposure
of the bare ice zone the following summer. This swift exposure would also include that of perennial rivers and lakes, much
earlier in the melt season than expected, hence the increased MF value observed in this study despite low MAR-derived melt
production and runoff values.

## 709    5.3 Satellite-derived MF and runoff simulations

We found a positive linear relationship between satellite-derived MF and MAR simulated R before runoff declined (Fig. 8),
showing how the MF-R relationship can be used to reliably simulate seasonal surface meltwater variation and provide further
understanding into how runoff is routed and stored, at least up to peak melt events. This finding supports other studies that
have used satellite-derived meltwater metrics and RCM-modelled runoff which have focussed on the southwest GrIS (Yang
et al., 2021), Northern GrIS (Lu et al., 2021; Li et al., 2022) and the Devon Ice Cap (Lu et al., 2020). In terms of post-peak
melt events, surface meltwater can still cover a substantially large area even after surface runoff has reduced and ceased, as
seen in this study. This is known as the 'delay' effect (Lu et al., 2021), whereby meltwater may continue to be routed or stored
on the ice surface via the slow routing of melt out of snowpack/firn at higher elevations, the stagnation and subsequent
preservation of transported meltwater in large supraglacial rivers or the storage of melt in SGLs. Therefore, whilst this MF-R
relationship is promising in providing comparative assessments between satellite observations and RCM-modelled runoff at
HG, calculated runoff volume via satellite- (Yang et al., 2021) or field-based measurements (Smith et al., 2017) are required
to provide further validation of such relationships, in particular over space and time across a full melt season.

**5.4 Future implications**

Substantial changes have taken place at HG over the last two decades driven by atmospheric and oceanic change (Carr et al.,
2015; Rignot et al., 2021). It is therefore important to consider HG and the overall northern regions sensitivity to such warming
under present climate scenarios. Northern Greenland is expected to undergo the greatest warming of the 21$^{st}$ Century across
the GrIS (Hill et al., 2017), and given its already low rates of winter accumulation compared to other ice sheet sectors (Goelzer
et al., 2013), means this region is likely to become ever more sensitive to climatic change in the future. Mapping performed
within this study illustrates the multi-annual persistence of the supraglacial drainage network within this high latitudinal region
(Fig. 4a) and the rapid and extensive response of this system to high melt years (Fig. 5). This response, in particular to the
extreme melt year of 2019, can precondition the ice surface for the following melt season, resulting in earlier but widespread
hydrologic activity and longer-lasting melt season. This preconditioning and subsequent behaviour is likely to become
increasingly normalised as melt events and atmospheric variability, such as persistent blocking events, increase in frequency
and intensity (Rahmstorf and Coumou, 2011; McLeod and Mote, 2016).
Inland migration of the supraglacial drainage network is also projected with continued warming (Leeson et al., 2015), with
recent work already showing ablation area expansion and amplification of melt and runoff post-1990 across Northern
Greenland (Noël et al., 2019). Many more surface depressions for future SGL locations are present above the current northern
ELA (Igneczi et al. 2016), with the potential to accumulate high volumes of meltwater and feed lengthening overflow
supraglacial rivers that extend tens of kilometres downstream to non-local, low elevation moulins (Poinar et al., 2015). For
ponded water, if ice becomes thin enough (Poinar et al., 2015), or localised ice columns become vulnerable to fracture from
refrozen ice complexities within firn (ice-blobs; Culberg et al., 2022) new hydrofracture events will bring such meltwater to
isolated areas of the ice sheet bed. This could have knock-on impacts to ice flow, with the likely delivery of water and heat to
a persistent inefficient subglacial system, where thicker, flatter ice may prohibit the development of an efficient subglacial
system (Dow et al., 2014), enhancing ice flow (Christoffersen et al., 2018).
Persistent low-permeability ice slabs which block vertical percolation have continued to thicken overtime in the lower
accumulation zone. Ice slabs are expected to enhance runoff from Greenland's interior, particularly in consecutive warm
summers (MacFerrin et al., 2019). Enhanced runoff and inland expansion of the supraglacial drainage network will impact
meltwater feedback processes, not only in driving overall SMB decline (Noël et al., 2021) but further impacting dynamical
behaviour, including hydrofracture potential. At HG, a particular concern is the vulnerability of its northern terminus to
increased hydrofracture events from greater melt runoff (Carr et al., 2015). Such events at HG have the potential to promote
future rapid run-away retreat of HG, especially if the northern sector of the terminus retreats beyond its pinning point into the
deepened bed (below sea level) in which is sits (Carr et al., 2015, Hillebrand et al., 2022).

As the northern region mainly consists of fast-flowing, marine-terminating outlet glaciers, like HG, that drain a large proportion
of the GrIS, further understanding of the mechanisms that drive their dynamical behaviour, in particular related to enhanced
runoff, are required for predicting their future contribution to GrIS mass loss and subsequent sea level rise.

## 6 Conclusion

In this study, we mapped and quantified for the first time, the spatial and temporal evolution of the supraglacial drainage
network, including both rivers and lakes, using 10 m Sentinel-2 images from the melt seasons of 2016 to 2020 at Humboldt
Glacier, northern Greenland. We identify an extensive supraglacial drainage network exists at Humboldt Glacier that is
particularly prevalent up to 1000 m a.s.l with a further variable transient zone extending up to ~1440 m a.s.l. The seasonal
evolutionary behaviour of this network migrates up-glacier in response to increasing runoff as air temperatures rise throughout
the melt season, with the network transforming from an inefficient system dominated by water percolation and slush flows to
one dominated by channelised, efficient flow. Interannual variability of the extent and behaviour of the system is associated
with high and low melt years across the study period, with the low melt years of 2017 and 2018 having both limited and
delayed spatial development. The extreme melt year of 2019 showed the extensive development and persistence of the
supraglacial drainage network into September, which followed by low snow accumulation during the subsequent winter/spring,
preconditioned the ice sheet for earlier hydrologic activity in 2020; behaviour which may become more representative with
extreme melt events and longer-lasting melt seasons into the future. This work ultimately contributes to advancing our
understanding of supraglacial hydrologic processes across the Greenland Ice Sheet by expanding detailed drainage mapping
to other understudied regions of the ice sheet, in particular to Greenland's rapidly changing northern region, aiding in
projections of future mass loss as enhanced runoff continues with climatic warming.

## 7 Data Availability

Supraglacial river and lake shapefiles can be requested by contacting the lead author. Regional climate model MAR v3.11 data
was provided by Dr Xavier Fettweis and is freely available via ftp://ftp.climato.be/ . Sentinel-2 imagery is available from the
Copernicus Open Access Hub (https://scihub.copernicus.eu) and digital elevation model ArcticDEM is available via
https://www.pgc.umn.edu/data/arcticdem/.     The automatic river detection algorithm is freely available via
https://github.com/njuRS/River_detection.

## 8 Supplementary Information

Supplementary data related to this article is available to view in the associated PDF.

## 9 Author Contributions

LDR designed the study, conducted data collection and analysis and prepared the manuscript. KY provided the source code for the automatic detection algorithm for automatic river mapping in Matlab. AJS aided with MAR data extraction for the Humboldt Glacier catchment. DMR, AJS, SJL and KY provided comments on draft versions of the manuscript produced by LDR.

## 10 Competing interests

Kang Yang is a member of the editorial board of The Cryosphere. The author(s) declare no other conflicts of interest.

## 11 Acknowledgements

LDR acknowledges financial support from a Natural Environmental Research Council (NERC) Doctoral Training Partnership (Grant number NE/L002450/1). Authors would like to acknowledge the Copernicus Open Access Hub for free access to the satellite imagery used in this study and also thank Dr Xavier Fettweis for the availability of MAR data. We also thank the Polar Geospatial Center for availability of the ArcticDEM. The authors would like to thank two anonymous reviewers for providing constructive comments which led to improvements of the manuscript and to the editor (Caroline Clason) for handling the manuscript.

## 12 Financial Support

This research has been supported by the Natural Environment Research Council (NERC) Doctoral Training Partnership (Grant number NE/L002450/1).

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
