# Peer review of "Seasonal evolution of the supraglacial drainage network at"

_The Cryosphere, 2023_

## Author Comment (AC1)

**Response to reviewer 1: 'Seasonal evolution of the supraglacial drainage network at Humboldt Glacier, North Greenland, between 2016 and 2020'. Rawlins *et al.* (2023)**

We would like the reviewer for their positive and insightful comments on our manuscript, which will lead to the following improvements. Our responses for each of the comments raised and how we addressed them are given below. Reviewer comments are italicised in blue with our responses in black. Please also find attached a revised manuscript with tracked changes. When referring to page numbers in the below text, these will be with page numbers associated on the revised (tracked change) manuscript.

*This is a super well written, clear, thoroughly referenced (rare!), and interesting paper. I don't get many of these to read, so this was a delight. I do have some suggestions below for how the paper could be improved, but they are all easily done and I think will make the paper stronger. I am very familiar with the subject matter and literature here, so my eyes glaze over on the methods sections and I took them for granted.*

We thank reviewer 1 for this positive overview and for their further constructive comments which are addressed accordingly.

*Slush definition- Slush zones are a key part of these results, but the authors have not described how they were defined/mapped/identified. I'd like to see their definition of a slush zone and how they uniquely identified them in images from other supraglacial features.*

Thank you for your comment on slush zones. Slush is defined as water-saturated snow when temperatures permit melting (above 0°C) which form a large expanse of surface pooled meltwater which can become mobile as slush flows or rill-type channels. Slush zones are of great hydrologic importance during the early melt season for initial meltwater mobilisation and later on in the season as headwaters, feeding the surface drainage network below and so were not extracted/treated independently. Slush is identifiable in true colour satellite imagery as a dense, light blue zone representative of shallow, water-saturated snow: distinguishable from the surrounding white colour of snow, grey colour of bare ice and darker, turquoise blue colour of supraglacial rivers and lakes (Holmes, 1955). An example of the slush zone is now provided as an extra panel in Figure 2 (g). In NDWI images, however, slush zones are more difficult to independently extract due to their similar spectral and linear shape signatures with other supraglacial hydrological components (i.e.., narrow rivers). Clarification of this is now included in the Methodology, lines 245 – 253 (revised manuscript).

*MAR uncertainty- I do not think the authors need an ensemble of models, but I would expect to see some discussion of the fact that MAR (or any coupled or uncoupled atmos-ice model) is our least bad representation of reality. There are known issues in this sort of modelling, and errors in the model will only strengthen your conclusions by potentially reducing scatter. Figure 5, for instance, should have a MAR uncertainty plotted on it, perhaps as a confidence interval. Discussion should also be added.*

We agree with this comment and have now added some initial discussions about the use of MAR, including its development for the study of polar regions (including Greenland), its inclusion of

important polar processes (e.g., SMB) and its evaluation against in-situ automatic weather station and satellite data sets (with reference to relevant literature). Further details regarding the high spatial (6 km) and temporal (daily) resolution of the MAR model version (v3.11) used is also given, as well as reference to literature where MAR data has been used in other supraglacial hydrologic studies. The inclusion of this text can be viewed in lines 201 - 211 in the revised manuscript.

We have also included uncertainty envelopes of MAR data used in original Fig. 5 (now Fig. 6 in revised manuscript), as recommended. This uncertainty is +/- 15% the MAR value calculated for the catchment (meltwater and runoff variables), as per Fettweis et al. (2020) and personal communication (Personal communication: Fettweis, 2023).

*MF uncertainty- as above, please discuss uncertainties in your MF. You have spatial errors resulting from pixel size, spatial errors from undetected sub-S2 channels, DEM resolution errors, and errors of classification that might omit/comit water area. This seems a larger omission from the paper- I (and readers) want to know where the method is good and where it is not. Since you aren't about to map a scene manually to provide true validation (although I wouldn't object to that!), I think you can propagate the variance from each of those terms based on the literature surrounding your classification methods. I think this information is essential to this paper.*

We agree with this comment and have now included a new figure (Fig. 3 in the revised manuscript) of a small mapped area of HG comparing both the automatic river detection algorithm vs manually digitised supraglacial rivers to assess performance accuracy. The results from this assessment show similarities between the overall spatial pattern of the automated and digitised networks, however small gaps are present in the automated rivers, which when quantified are 5.4% shorter than those which are manually digitised. This new section of writing and figure can be found in the Methodology, Section 3.2, lines 258 – 262 (in the revised manuscript). We believe this provides the information required for MF uncertainty in this paper.

*Drainage density and other stats- you have the data to create a vector river network and determine drainage density, stream orders, and other stats for comparison (see last point). Author Yang has published many papers on this topic and therefore I believe this should be a straightforward task that would add needed richness to this paper in terms of comparison.*

Whilst Drainage density ($D_d$) has been used previously in other publications, this stat was not originally included within the manuscript as such results were comparable with the use of the Meltwater Area Fraction (MF; a stat introduced and used by Lu et al. (2021) and Yang et al. (2021)), with behaviour of both statistics being connected (which would lead to repetition). Instead, we have now included $D_d$ statistics in Table S2 (Supplementary Information) and have attached graphs of $D_d$ and MAR-derived values (see Fig. R1: similar to that of Fig. 6 in the revised manuscript) to show their paralleled behaviour.

[Figure]

**Figure R1**: *Drainage Density ($D_d$, km$^{-1}$) and MAR-derived meltwater production and runoff (mm w.e. day$^{-1}$) for each mapped study date across the study period (2016-2020) for the HG catchment, north Greenland. +/- 15% uncertainty envelopes are provided for MAR-derived values (Fettweis et al., 2020).*

In regards to other hydrologic statistics (e.g., stream order, braiding index), this is beyond the scope of this paper, which focusses on mapping the supraglacial hydrologic network (both rivers and lakes) in an unmapped location at high spatial and temporal resolution, and to do this over a number of consecutive years (for the first time) to assess the seasonal behaviour. We believe that such in-depth, targeted hydrologic statistics (e.g, drainage density, stream order) would be well placed in an additional, future paper focussing on only melt year, date or internally drained catchment of HG to enable a more thorough investigation into such metrics (such as those performed in Smith et al., 2015; 2017; Gleason et al., 2021; Muthyala et al., 2022).

Gleason, C.J., Yang, K., Feng, D., Smith, L.C., Liu, K., Pitcher, L.H., Chu, V.W., Cooper, M.G., Overstreet, B.T., Rennermalm, A.K. and Ryan, J.C.: Hourly surface meltwater routing for a Greenlandic supraglacial catchment across hillslopes and through a dense topological channel network. The Cryosphere, 15(5), pp.2315-2331, https://doi.org/10.5194/tc-2020-273, 2021.

Muthyala, R., Rennermalm, Å. K., Leidman, S. Z., Cooper, M. G., Cooley, S. W., Smith, L. C., and van As, D.: Supraglacial streamflow and meteorological drivers from southwest Greenland, The Cryosphere, 16, 2245–2263, https://doi.org/10.5194/tc-16-2245-2022, 2022.

Smith, L.C., Chu, V.W., Yang, K., Gleason, C.J., Pitcher, L.H., Rennermalm, A.K., Legleiter, C.J., Behar, A.E., Overstreet, B.T., Moustafa, S.E. and Tedesco, M.: Efficient meltwater drainage through supraglacial streams and rivers on the southwest Greenland ice sheet. Proceedings of the National Academy of Sciences, 112(4), pp.1001-1006, https://doi.org/10.1073/pnas.1413024112, 2015

Smith, L. C., Yang, K., Pitcher, L. H., Overstreet, B. T., Chu, V. W., Rennermalm, Å. K., et al.: Direct measurements of meltwater runoff on the Greenland Ice Sheet surface. Proceedings of the National

*Academy of Sciences of the United States of America, 114(50), E10622–E10631. https://doi.org/10.1073/pnas.1707743114, 2017.*

*Split MF into river and lake areas- I am quite interested in this divide. This is figure S3, but for me this belongs in the main text as a very interesting expression of the supraglacial hydrology here. Some discussion should also occur.*

We agree with this comment and Figure S3 has now been moved into the main results as Fig. 7 with associated text (457 – 468, revised manuscript) and additional discussion included (629 – 647, revised manuscript), including reference to the difference in percentage composition (i.e., rivers and SGLs) of the drainage networks between the north and southwest regions of Greenland.

*Section 5.4- I am ok with this section as it is fairly hedged and well referenced, but other reviewers may not like to see such speculative conjecture.*

We thank the reviewer for their comment.

*Missing comparisons- the 2nd major omission I see (beyond uncertainty discussion noted above) is a lack of comparison to the rich literature of the SW GrIS. This is the right paper to use the discussion to first outline this bit of ice sheet (as you have done) and then explicitly compare to the SW to see what is the same and what is different- are the fractions of rivers and lakes the same? Elevations of highest melt features? Density of persistent channels? Channel lengths? Width distributions of these channels? Prevalence of slush zones/bare ice and their interaction with the network? I think you have all the data to answer those questions (and more) and I think this paper really needs it to move this beyond an interesting and well written observational study into a richer contextual understanding of this unique bit of ice. I'd like to see these differences quantified where possible (e.g. from a vector network) or discussed qualitatively and referenced where not possible (as you have nicely done throughout this paper!!)*

We have now incorporated a wider comparison of our data with studies from southwest/west Greenland throughout the discussion. This includes a comparison between the maximum elevation of the supraglacial drainage network, which is ~400 m a.s.l lower (1440 m a.s.l) at HG than in southwest Greenland (~1800 m a.s.l; Yang et al., 2021): a finding consistent with another mapping study by Lu et al. (2021) from northeast Greenland. Other comparisons are also made, including differences between the parallel-style of drainage pattern mapped in the southern sector of HG and the typical dendritic-style of drainage observed in southwest Greenland. Additionally, the composition of the drainage network differs, with supraglacial rivers contributing significantly more to the network at HG than southwest Greenland, where SGLs and associated meltwater storage is higher. This increased contribution of supraglacial rivers in the northern network is attributed to the reduced presence of available depressions for SGL development, steeper slopes promoting runoff over storage and the inclusion of slush zones within this study, which likely increases the MF and RF value. We do, however, express caution in direct comparison of metrics between regions as our study uses a lower threshold ($t_5$) and retains slush zones, which will increase river components (lines 639 – 647 in the revised manuscript). Some similarities in drainage between the two regions are also considered, including the

lengthening of channels to higher elevations from the already-existing network. We thank you for your comment on the inclusion of this important aspect.

---

## Author Comment (AC2)

**Response to reviewer 2: 'Seasonal evolution of the supraglacial drainage network at Humboldt Glacier, North Greenland, between 2016 and 2020'. Rawlins *et al.* (2023)**

We would like to thank the reviewer for their positive and insightful comments on our manuscript, which will lead to improvements. Our responses for each of the comments raised and how we addressed them are given below. Reviewer comments are italicised in blue with our responses in black. Please also find attached a revised manuscript with tracked changes. When referring to page numbers in the below text, these will be with page numbers associated with the revised (tracked change) manuscript.

*This paper was a pleasure to read as it is written so well. I only have some minor comments (below), followed by minor specific line by line comments.*

We thank the reviewer for their positive overall comments and for their constructive points below which have been addressed accordingly.

*The incorporation of slush mapping is very interesting and novel, as I am not aware of prior studies that have mapped slush in detail Greenland(?), though this has been done by at least one study on Antarctic ice shelves (e.g. Dell et al 2022). So it would be great to see more details regarding how exactly slush was mapped in this study. Additionally, I think slush (e.g. as a component of total meltwater area fraction (MF)), could feature more heavily as a discussion point in the Results/Discussion.*

Thank you for your comment on slush zones. Slush is defined as water-saturated snow when temperatures permit melting (above 0°C) which form a large expanse of surface ponded meltwater that can become mobile as slush flows or within rill-type channels. Slush zones are of great hydrologic importance for the initial mobilisation of surface meltwater at the beginning of the melt season and its migration upglacier to act as headwaters feeding the drainage network below as the melt season progresses, hence why they were retained within the mapped hydrologic network and not treated independently. Slush is identifiable in true colour satellite imagery as a dense, light blue zone representative of this shallow, water-saturated snow layer: distinguishable from the surrounding colour of snow (white), darker bare ice (grey) and other supraglacial rivers and lakes (turquoise blue: Holmes, 1955). An example of this slush zone is now provided as an extra panel in Fig. 2 (g). Additionally, in contrast to true colour images, in NDWI images, slush zones are more difficult to independently extract due to their similar spectral and linear shape signatures with other drainage components (i.e., rivers). This clarification is now added to the Methodology, lines 245 – 253.

*I do not know Humboldt Glacier well, so I wondering how much of this marine terminating glacier is floating? Unless I missed it, the only mention of floating ice is on line 208, where a '7 km floating section' is mentioned. Depending on the area of this floating area of ice, I'd also be particularly interested to know whether the authors see a difference between lake/other meltwater feature characteristics on the floating versus grounded portions of the glacier? (For example, comparisons of meltwater features on floating vs. grounded ice have previously been made on Peterman Glacier*

*(Macdonald et al 218) and well as Paakitsoq (SW Greenland) vs. Larsen B Ice shelf, Antarctica (Banwell et al 2014).*

A previous study by Carr et al. (2015) stated that within 25 km of the northern sectors calving front, HG is heavily crevassed (including water-filled), with the lower 6.5 km near floatation, producing large tabular icebergs. In this study, we avoid this area due to the potential for erroneous delineation of crevasses instead of rivers in the lower reaches of Humboldt Glacier (in particular the northern sector), due to their similar spectral characteristics. Therefore, the lower crevassed portion of Humboldt Glacier was removed using a manually-created crevasse mask to avoid impacts (i.e., overestimation) of calculated metrics of MF in these lower elevations. We are therefore unfortunately unable to provide further conjecture on floating vs grounded portion of HG, however appreciate the wider context this may have brought and an important consideration for future work. The use of the crevasse mask and why is stated in Section 3.2 (lines 226 - 233).

*I am wondering why the authors choose to base the NDWI on the Green and NIR bands rather than the blue and red bands, e.g. as used by Bell et al (2017) and Williamson et al (2018). I am sure there were/are good reasons, but perhaps a sentence or two about this could be added to the paper. Also related to the use of the NDWI, I am wondering how/why the authors decided to use a threshold of 0.4, i.e. which they call a 'high-value global NDWI threshold'? Again, I am not suggesting this threshold is not appropriate, but perhaps some more detail and/or reference(s) be added about this choice of value?*

The use of the NDWI (McFeeters, 1996) was used rather than NDWI$_{ice}$ due to the preferential inclusion of additional shallow meltwater characteristics, including slush zones, which NDWI$_{ice}$ was originally developed to reduce in order to produce fewer false classifications of water over blue ice and slush areas (see Yang and Smith, 2013). Also, a preliminary performance accuracy test in a sample area which examined their differences found NDWI (McFeeters, 1996) performed more regularly connected river channels than NDWI$_{ice}$ by 16.8%. We have now included a sentence about this in lines 221 – 226 (revised manuscript) and have included an additional figure in the Supplementary information (Fig S1) to show this.

In regards to the NDWI threshold of 0.4, this was chosen as best captured the boundaries of SGLs and wide supraglacial river segments. This threshold was also based on other literature (Lu et al., 2021) which has now been included as a citation here.

*My final general comment is that I think the authors should add a short paragraph about uncertainty quantification, particularly regarding their MF analysis.*

We have now included a new figure (Fig. 3 in the revised manuscript) of a small area of HG comparing both rivers derived from the automatic river detection algorithm and those which are manually digitised to assess performance accuracy. The results from this assessment show similarities between the overall spatial pattern mapped between both automated and manually digitised networks, however small gaps are present within the automated rivers, which quantify as 5.4% shorter than the manually digitised. This new section of writing and figure can be found in Methodology, Section 3.2, lines 258 – 262 (in the revised manuscript).

*Abstract lines 16 and 17: Not necessary to state areas to 1 d.p. Round up as is done for other area/elevation values in the abstract.*

Amended.

*Abstract line 19: I suggest adding an few extra words to explain what you mean by 'preconditioning' here.*

Extra words have been added to the abstract.

*56 – 57: Two relevant studies focusing on the surface hydrology of Petermann Glacier in NW Greenland could also be referenced here: Macdonald et al (2018) and Boghosian et al (2021).*

Additional references added.

*133: Gledhill and Williamson just have a 2017 paper, but Williamson et al. have a 2018 paper (both are already in the reference list).*

This has now been altered. Confusion in the original manuscript was between Gledhill and Williamson (2018) and Williamson et al. (2017; 2018). Thank you for raising this.

*241 – 247: I find this paragraph confusing regarding the maximum extent of meltwater mapped, versus the maximum extent of the study region. For example, the first sentence says that rivers and lakes were mapped up to a "maximum melt extent of 1500 m a.s.l.". Is 1500 m the highest elevation of the study area analyzed, or is this the maximum elevation of observed melt? I assume the latter(?), but if so, then the following sentence is repetitive (i.e. this states "The mapped supraglacial drainage network across HG is shown extend up to 1500 m a.s.l,"). Also, later in the paragraph (line 246), it says rivers and lakes from up to a "maximum of 1440 m" (as opposed to 1500 m). So these sentences need to be re-written for clarity.*

Maximum extent of meltwater mapped has now been clarified (lines 277 – 280 in the revised manuscript). Thank you.

*253 - 255: For the sentence "In Figure 3b, we also see some evidence of a potential main-river reconfigurations, with the north-westward advection of a river channel that runs transverse to ice flow"; maybe this river in Fig 3b could be labelled? As I see various rivers/streams that are transverse to ice flow. Also, it looks to me as though similar examples may also be seen in panels c and e?*

We have now identified the potential reconfigurations in Fig. 3b (now Fig. 4b in the revised manuscript), as well as panel e where this is also visible by red arrows.

*310 – 314: Can the authors suggest a possible explanation for why these two parallel lines that track across glacier exist? Could they be fractures?*

We believe these two parallel lines are depressions within the ice surface with potential fractures (i.e., moulins) associated with them due to the abrupt termination of main river channels here. We have included additional clarification about this in lines 338 - 341.

*497 - 499: For the sentence:"… this study also notes that many well-established rivers that are longitudinal to ice flow, including many with canyonised features, also reoccupy locations.", studies focused on Petermann Glacier could also be mentioned here, which found similar findings I believe (Macdonald et al. 2018, Boghosian et al. 2021).*

Studies have now been included within the citation (and reference list).

*622/623: Mention Summer 2019 somewhere in this sentence to remind the reader which melt season is being described.*

Summer 2019 added.

*Fig 2: it would be interesting to know the locations of these figure panels, so perhaps an extra panel could be added to show this (e.g. as is done in Fig 3a), or perhaps the locations should be shown somewhere in Fig 1? Also, the 'off edge river termination' feature in panel 2c) is interesting, and I'm wondering how comparable this feature could be to the large river/waterfall described in Bell et al (2017)?*

Thank you for this recommendation. We have now included an extra panel in this figure (Fig. 2 in the revised manuscript) of the locations for subsequent images for reference purposes. An extra panel (Fig. 2g) has also been added as an example of a slush zone on HG.

*References (those in bold are not referenced in the current paper)*

*Banwell, A.F., Cabellero, M., Arnold, N., Glasser, N., Cathles, L.M., MacAyeal, D. 2014. Supraglacial lakes on the Larsen B Ice Shelf, Antarctica, and Paakitsoq Region, Greenland: a comparative study. Annals of Glaciology. 55(66), doi:10.3189/2014AoG66A049.*

*Bell, R. E., Chu, W., Kingslake, J., Das, I., Tedesco, M., Tinto, K. J., Zappa, C. J., Frezzotti, M., Boghosian, A., and Lee, W. S.: Antarctic ice shelf potentially stabilized by export of meltwater in surface river, Nature, 544, 344–348, 2017.*

*Boghosian, A.L., Pitcher, L.H., Smith, L.C. et al. Development of ice-shelf estuaries promotes fractures and calving. Nature Geoscience, 14, 899–905 (2021). https://doi.org/10.1038/s41561-021-00837-7*

*Dell, R., Banwell. A.F., Willis, I., Arnold, N., Halberstadt, A.R.W., Chudley, T.R., Pritchard, H. 2022, Supervised classification of slush and ponded water on Antarctic ice shelves using Landsat 8 imagery, Journal of Glaciology, 1–14. https://doi.org/10.1017/ jog.2021.114.*

*Gledhill, L. A. and Williamson, A. G.: Inland advance of supraglacial lakes in north-west Greenland under recent climatic warming, Annals of Glaciology, 59, 66-82, https://doi.org/10.1017/aog.2017.31, 2018.*

*Macdonald, G.J., Banwell, A.F., MacAyeal, D.R. 2018, Seasonal evolution of supraglacial lakes on a floating ice tongue, Petermann Glacier, Greenland. Annals of Glaciology, doi:10.1017/aog.2018.9*

*Williamson, A. G., Banwell, A. F., Willis, I. C., and Arnold, N. S.: Dual-satellite (Sentinel-2 and Landsat 8) remote sensing of supraglacial lakes in Greenland, The Cryosphere, 12, 3045-3065, https://doi.org/10.5194/tc-12-3045-2018, 2018.*

These references have now been included as citations (where appropriate) and the reference list. Thank you.